# A game-theoretic analysis of Wikipedia's peer production: The interplay between community's governance and contributors' interactions

**Santhanakrishnan Anand**[1]\*, **Ofer Arazy**[2], **Narayan Mandayam**[3], **Oded Nov**[4]

**1** Department of Electrical and Computer Engineering, New York Institute of Technology, New York, NY, United States of America, **2** Department of Information Systems, University of Haifa, Haifa, Israel, **3** Wireless Information Networks Lab (WINLAB), Department of Electrical and Computer Engineering, Rutgers University, North Brunswick, New Jersey, United States of America, **4** Department of Technology Management, New York University, Brooklyn, New York, United States of America

\* asanthan@nyit.edu

**Data Availability Statement:** All relevant data are within the paper and its Supporting information files.

## Abstract

Peer production, such as the collaborative authoring of Wikipedia articles, involves both cooperation and competition between contributors. Cooperatively, Wikipedia's contributors attempt to create high-quality articles, and at the same time, they compete to align Wikipedia articles with their personal perspectives and "take ownership" of the article. This process is governed collectively by the community, which works to ensure the neutrality of the content. We study the interplay between individuals' cooperation and competition, considering the community's endeavor to ensure a neutral point of view (NPOV) on articles. We develop a two-level game-theoretic model: the first level models the interactions between individual contributors who seek both cooperative and competitive goals and the second level models governance of co-production as a Stackelberg (leader-follower) game between contributors and the communal neutrality-enforcing mechanisms. We present our model's predictions regarding the relationship between contributors' personal benefits of content ownership and their characteristics, namely their cooperative/competitive orientation and their activity profile (whether *creators* or *curators* of content). We validate the model's prediction through an empirical analysis, by studying the interactions of 219,811 distinct contributors that co-produced 864 Wikipedia articles over a decade. The analysis and empirical results suggest that the factor that determines who ends up owning content is the ratio between one's cooperative/competitive orientation (estimated based on whether a core or peripheral community member) and the contributor's creator/curator activity profile (proxied through average edit size per sentence). Namely, under the governance mechanisms, the fractional content that is eventually owned by a contributor is higher for curators that have a competitive orientation. Although neutrality-seeking mechanisms are essential for ensuring that ownership is not concentrated within a small number of contributors, our findings suggest that the burden of excessive governance may deter contributors from participating, and thus indirectly curtail the peer production of high-quality articles.

**Funding:** This work was partially funded by the United States National Academies Keck Futures Initiative (NAKFI).

**Competing interests:** The authors have declared that no competing interests exist.

## 1 Introduction

Over the past three decades, peer-produced online information goods such as open source software and Wikipedia have become extremely popular. Wikipedia recruited over 30 million volunteers to produce hundreds of millions of encyclopedic entries in 314 languages. As Wikipedia has become one of the most popular information sources on the web and the destination most internet users turn to when they seek information [1], the quality of its articles has been the topic of public debate. Wikipedia is based on wiki technology: a web-based collaborative authoring tool that allows contributors to add new content, append existing content, and delete or overwrite prior contributions [2]. Thus, Wikipedia articles evolve through the continuous additions, deletions, and shaping of existing content (rewriting, reorganizing, and integrating).

The success of online production communities, such as Wikipedia, has attracted much interest within the scholarly community, given that the organization of such communities seem to defy traditional organizational logic [3–7]. Broadly speaking, two key issues have been investigated. The first pertains to contributors' characteristics and their interactions during the collaborative production [8–12]. Namely, given that most communities do not have formal procedures to select, train and assign contributors to a task, it is not clear how work effectively gets done and how high-quality goods are co-produced. This issue is particularly interesting since communities are often heterogeneous and members vary greatly in terms of their motivation, affiliation with the community and identification with its goals, and their activity profiles. The second key issue is community governance [13–15]. Specifically, communities need to balance openness and freedom of action so as to allow contributors to self-select and self-organize, while controlling uncooperative behavior, such as manipulations and vandalism. In other words, there is a fundamental tension between emergent behavior and regulation in peer production [16]. Successful communities, thus, had to devise "light" governance mechanism [15] that would strike a fine balance between these two opposing objectives. Extensive research in the area over the last three decades has been able to shed light on many aspects of peer production; still, fundamental questions remain unclear, specifically regarding the way in which governance shapes individual-level interactions. Hence, the overarching objective of our study is to investigate the interplay between the community's governance and contributors' cooperation-competition dynamics in peer production (and specifically, in Wikipedia).

Large and complex socio-technical systems, such as Wikipedia and other large-scale peer production projects, are difficult to study in a holistic manner, and by and large, scholarly investigations have focused on a particular angle, for example contributors' motivations [11, 17], coordination [9], or conflict management [8]. One particularly relevant approach for studying such complex systems is through modeling. A model, by definition, presents a simplified depiction of the complex reality, formalizing key salient features of the system, while ignoring other less critical aspects. Models have the potential to reveal deep insights about the phenomenon being investigated. Moreover, when the model-based predictions are validated with empirical data, credence is lent to the model's results. In this study we employ game-theory to study the interplay between the community's governance and contributors' individual-level interactions.

Game theory, "the study of mathematical models of conflict and cooperation between intelligent rational decision-makers" [18], is well suited for studying competitive and cooperative dynamics and thus may reveal insights into peer production. Despite its demonstrated utility in studying cooperation and competition, the use of game theory in peer production research has been rather limited [19]. In this study, we employ game-theoretic modeling in order to study the interplay between the community's governance and contributors' individual-level

interactions. More specifically, we develop a two-step leader-follower game-theoretic model [20, 21] that captures both cooperation-competition dynamics among the contributors to the article's collective authorship (i.e., "follower"), as well as the community's governance efforts to enforce neutrality (i.e., "leader"). We briefly detail the key elements of our model, beginning with the former: the dynamics of contributors' interactions.

The "follower" level of our model assumes a heterogeneous community, where contributors are characterized by two important traits. First, we assume that each contributor combines self-interest (e.g., promoting one's viewpoints and enhancing their own reputation [22, 23]) with others-oriented interests (e.g., helping the community co-create high-quality goods) [24]. The self-interest is linked to a competitive behavior and others-orientation to a more coopera- tive conduct [25]. A contributor is driven by a combination of cooperative and competitive goals, and we refer to this trait as the contributor's cooperative/competitive orientation.

A second important characteristic that we model is contributors' activity profile. Despite the freedom to select the particular ways in which one contributes (for example, in Wikipedia: adding new content or shaping others' previous contributions), members of online production communities are typically characterized by highly stable activity patterns (aka "emergent roles"; [26]). Building on prior studies in the area [27, 28], we distinguish between the roles of a creator (in Wikipedia, mostly adding new content) and a curator (refining, shaping and reor- ganizing existing content), modeling each individual's activity profile as some combination of creation and curation activities. It should be noted that in the context of Wikipedia, the crea- tor/curator distinction applies to activity in the primary area (referred to as the Main name- space): co-producing encyclopedic entries; other areas (namespaces) within Wikipedia are dedicated to coordination work. For example, each article is coupled with a TalkPage where discussions regarding the article take place. Our model assumes that these two characteristics, i.e., contributors' cooperative/competitive orientation and their creator/curator profile are sta- ble traits (at least within the timeframe of our analysis).

We further presume that the contributor's decision variable is the the amount of work to be done (i.e., number of content elements, e.g., sentences, to be edited), reflecting their individual motivation and availability. This decision is influenced by the utility function, i.e., the balance of costs and benefits. We assume that contributors derive both competitive benefits (namely, shaping a Wikipedia article to align with one's personal viewpoint, thus coming to "own" por- tions of that article) and cooperative benefits (i.e., co-producing a comprehensive encyclopedic entry). Similarly, the costs of editing the focal article are influenced by one's activity profile (creator/curator, and hence are applicable to both cooperative and competitive contributors), the amount of coordination work required (more applicable to cooperatively-oriented contrib- utors) and the effort of handling the community's governance (more applicable to competi- tively-oriented contributors). When many contributors participating in a collective authoring effort attempt to promote their reputation and perspectives, competition between them is inevitable [25].

Contributors' dynamics does not occur in a void. The interactions among the contributors are influenced by the community's governance (i.e., the "leader" in the two-stage game-theo- retic model). Wikipedia's governance mechanisms have been investigated in prior studies, e.g., [13–15]. These studies describe Wikipedia's extensive mechanisms for ensuring that the co- produced content is of high quality, including instruments such as norms, policies, as well as technical tools for facilitating collaboration, resolving conflicts and fighting vandalism [29]. In particular, Wikipedia emphasizes objectivity (Neutral Point of View; NPOV) as a central pillar of the community (http://en.wikipedia.org/wiki/Wikipedia:FivePillars) and lists "susceptibility to editorial and systemic bias" as one of the key aspects of its quality assurance work [25, 30, 31], http://en.wikipedia.org/wiki/ReliabilityofWikipedia. Of particular relevance to our

investigation are studies of governance mechanisms that are intended to ensure neutrality by countering attempts toward bias in Wikipedia articles [32, 33].

To recap, contributors are characterized by their cooperative/competitive orientation and by their typical activity pattern (creators or curators of content). A contributor's decision regarding the amount of content to be edited is based on that individual's utility function, and is also influenced by the strategies used by other contributors, which in turn are affected by adherence to governance (i.e., the principles of neutrality). Contributors' optimal strategies are calculated using the *Nash equilibrium* of the game.

To lend greater credence to our model's predictions, we validated the findings with empirical data. We performed a computationally-heavy empirical analysis of 219811 distinct contributors co-producing 864 Wikipedia articles (articles' lifespan ranges from 129 to 4078 days; mean = 2681 days). We performed the empirical validation for the entire period covered by our data set (2001–2012), as well as for each yearly interval independently. We estimated contributors' cooperative/competitive orientation based on their role in the community (those closer to the community's core are assumed to be more cooperative) and estimated their position on the creator-curator continuum based on the average size of their edits (i.e., larger edits within a sentence are typical of content creators) [34–37].

To verify the validity of the "creator"/"curator" labels, we performed an analysis on the dataset from [38], calculating each contributor's average edit size and profiling the types of activities that she makes. We found that: (a) the larger the relative proportion of the "Add Content" and "Structural Changes" edit types in a contributor's profile, the larger is the contributor's average edit size; and (b) the larger the proportion of "Corrections" and "Add References" edit types, the smaller is the contributor's average edit size. Together, these patterns suggest that contributors with a large edit size are the "creators" who structure and article and add much of the content, whereas those making small edit size are "curators" that mostly engage in refining existing content.

We used a sentence as our basic content unit, and our estimate of the cost of editing a sentence, i.e., the effort contributors invest, is approximated as a linear function of contributors' average edit size per sentence [39, 40]. We estimated contributors' competitive benefit based on the portion of the article that they own, employing the "content ownership" algorithm from a prior study [38], and their cooperative benefit from the comprehensiveness of the co-produced article (estimated through the article's length).

The key results of our game-theoretic analysis are (*a*) the factors that determine who ends up owning content are contributor's characteristics: their cooperative/competitive orientation and their creator/curator activity profile, and (*b*) governance levels should be limited, so as not to overburden contributors.

## 2 Background and related work

This section reviews prior work on Wikipedia, addressing the following issues: (a) contributors' cooperation and competition (Section 2.1); (b) the activities involved in Wikipedia's collaborative authoring process (Section 2.2); and (c) community-based governance of Wikipedia's peer production (Section 2.3).

### 2.1 Cooperation and competition in Wikipedia's content production

Scholarly accounts of peer production initiatives such as Wikipedia tend to emphasize the collaborative aspects [3, 8, 10, 41, 42]. Namely, contributors collaborate in the creation of encyclopedic entries, collectively attempting to ensure the quality of these articles. Cooperatively, participants in Wikipedia's collaborative-authoring process attempt to create encyclopedic

entries that cover the article's topic in a comprehensive manner, are factually accurate, as well as are: objective (i.e. unbiased), timely and relevant [43, 44]. Particularly, comprehensiveness is considered a key indicator of information quality [45], for example, the number of words or sentences is a very good predictor of article quality [46, 47]. Hence, cooperative Wikipedia members contribute their knowledge and share content, with the goal of collectively covering the focal topic in a comprehensive manner. Cooperation also entails the investment of additional effort beyond the direct work of peer-producing articles. Namely, collaborative contributors often take on additional roles and responsibilities, such as the coordination of work and in the enforcement of policies. The cooperative goals are well understood, and thus are only described here in brief. However, participants are also driven by competitive goals [48] and explaining these goals requires some elaboration.

Given that Wikipedia has become the primary entry point for those seeking information on the web [1], the impact, and thus the associated benefit, of shaping an article's contents can be high [49], as demonstrated by commercially sponsored attempts to bias Wikipedia's content [22, 23]. Recent news stories revealed that commercial entities-often through agents, such as PR agencies or political operatives-have been attempting to manipulate Wikipedia articles' content [23, 50, 51]. Such manipulations are intended to portray the interested parties favorably, and have been carried out over years, often employing sophisticated methods (e.g., using multiple accounts, building trust in the community and gaining access to special privileges) [52]. Wikipedia fights these manipulations through social, technical and legal means [22].

Additionally, competition over content can also originate from well-intentioned contributors. Prior studies have looked at conflicts of opinion between contributors [8], focusing on conflict resolution mechanisms and the impact of conflicts on the article's quality. More recently, several research studies explicitly discussed contributors' attempt to influence articles. For example, [53–55] describe various forms of biases and violations of Wikipedia's NPOV policy, and propose automatic methods for detecting these biases. Moreover, recent studies have demonstrated empirically that Wikipedia contributors often shape an article in a way that is aligned with their worldview, so that the articles often evolve through the "pulling" of the article towards the direction that represents the editor's perspective. [56] stated that "contributors to a focal artifact manipulate the article according to their particular viewpoints, thus pulling the artifact's trajectory in different directions" (p. 2014) and then suggested that such "pulls" may possibly stem from "· · · a more deliberate attempt to shape the artifact according to the contributor's personal vision." (p. 2016). Together, these studies demonstrate that competition and struggle over the views expressed in articles are salient features of Wikipedia's co-production.

Psychological ownership theory [57] may provide a potential explanation for why contributors compete to own pieces of an article. According to this theoretical framework, actors that occupy a shared social space can easily form an ownership feeling over a target (in this case, a Wikipedia article) if they invest much time or energy on it, are familiar with it, or have control over it. This is particularly pertinent in cases where the individual is the originator of the target (or portion of that target), as in the case of contributing content to a Wikipedia article. A contributor of content to the article will feel the content is her personal psychological property, and subsequently will be unwilling to either lose control of or share the ownership with others [58].

Territoriality theory [59] may further explain how psychological ownership could result in competition over an article's contents. According to this theory, the stronger an individual's sense psychological ownership of an object, the greater the likelihood they will treat that object as their territory. Territoriality behavior is likely to emerge as community members work together to accomplish a common goal. It signals to others one's stake in a territory or an

object. Territoriality may be expressed through the defense of one's turf from perceived invasions [59]. From such a viewpoint, anyone experiencing a strong sense of ownership of personally contributed content may exhibit territorial behavior over that section of the article, which subsequently leads one to defend the "owned" content from others' attempts to change or overwrite it [60–63]. Contributors may also try to protect a particular version of the article and exhibit territorial behavior even if they did not edit the article, e.g., by protecting the content of like-minded contributors. Kane et al [64] reported that roughly two-thirds of the collaborative-authoring patterns in the Wikipedia article they had investigated entailed "defensive filtering". They explained that in the defensive filtering pattern, "The focus of production was on protecting the content that has been created by the co-production community" and that the nature of the interaction was defensive and combative. Contributors may become committed to stabilizing knowledge already co-produced. For example, when assessing whether a change to the article fits with the article's current articulation, they may try to ensure that it does not "violate organizational and content decisions already made" [64]. [65] describe power dynamics in Wikipedia whereby certain contributors may attempt to dominate an article and block alternative viewpoints. [33] described such a domination dynamic that was related to gender bias, where core contributors wrestled against peripheral contributors to shape articles according to their viewpoints.

While territoriality may be linked to positive behaviors, such as increased motivation and commitment and thus could help in the organization of work, it raises concerns regarding competitive tensions and the potential for biases in the co-produced articles. For example, [60] showed that territorial behavior is directly linked to turf wars (namely, the revert of a Wikipedia article to its previous version). Furthermore, territorial behavior may deter members from participating [63].

The scholarly literature in the field shows that people choose to play different functional roles within an online community [66], broadly organized into a core-periphery structure [67], where the small core includes highly-committed individuals who take on various administrative duties and work to serve the community (i.e., cooperative orientation), and a much larger periphery that includes less-committed contributors who often are interested in promoting personal interests (i.e., competitive orientation) [24]. This community structure has been shown to be mostly consistent, as contributors keep their core/periphery positions over prolonged periods [66], although over extended time, some contributors may gradually transition from the community's periphery to its core [68].

## 2.2 Contributors' activity profiles: Creators and curators

To understand contributors' competition over content ownership, it is important that we explicate what it means for a contributor to "own" a unit of content, say a sentence. Ownership could be gained by either originating the content (i.e. adding substantive content) or alternatively by shaping existing content (reorganizing, completing missing pieces, adding references, etc.). For example, the content ownership algorithm that was employed by [38] assigns the ownership over a sentence to the contributor who has added/changed over 50% of the words in that sentence. A contributor to Wikipedia may engage in the creation of new content or in the curation of content that was originated by others.

Several studies have attempted to profile the activity patterns of Wikipedia's contributors, and their empirical results are quite consistent [26–28]. Broadly speaking, these studies have identified several prototypical activity patterns that fall into two primary categories: (a) creators of content, e.g., "substantive expert" in [28] or "quick-and-dirty editors" in [26] or "starters" in [27]; (b) curators of content, responsible for the shaping of content and copyediting,

e.g. "fact-checker" and "copyeditor" in [28], "content shapers," "layout shapers" and "copyedi-tors" in [26], and "content justifiers," "cleaners" and "copyeditors" in [27]. Interestingly, these studies have also identified a hybrid profile that is engaged in both creation and the curation of content, e.g. "all-round-contributors" in both [26, 27] This suggests that creation and curation are not exclusive, such that a contributor's activity profile may be positioned anywhere on the creation-curation continuum.

It is worth mentioning that studies of Wikipedia have also identified activity profiles that do not directly influence the content of Wikipedia articles. For example, vandals and vandal-fighters (e.g., "watchdogs"), which represent only a small portion ($\sim$ 8%) of Wikipedia's edit-ing activity [56]. Vandalism does not have much influence on the content that is made avail-able to readers, as it is very quickly corrected such that it doesn't have a lasting effect on Wikipedia articles [69, 70]. Hence, the treatment of vandalism is left outside the scope of the current study. In addition, some Wikipedians are engaged in administrative tasks and in the governance of the community, being active primarily in auxiliary areas of Wikipedia, and this activity does not directly affect the main article wiki pages. Hence, these contributor profiles are less relevant for the current study.

## 2.3 Governance of Wikipedia's content production: Neutral Point of View (NPOV) policy

Distributed collaborative communities often employ policies and community norms in order to prevent territorial behavior. The Wikipedia community explicitly discourages contributors from taking ownership of their work. Wikipedia founder, Jimmy Wales, speaks openly against the notion of ownership. For example, see a talk in the 21st Chaos Communication Congress in December 2004 (talk titled "Wikipedia Sociographics"); http://ccc.de/congress/2004/fahrplan/event/59.en.html. Also, the "Ownership of Content" policy (see https://en.wikipedia.org/wiki/Wikipedia:Ownership_of_content retrieved February 8, 2021) specifically points out that being a primary contributor is not a reason for asserting possession of an article:

> "*All Wikipedia content – articles, categories, templates, and other types of pages – are edited collaboratively. **No one** [emphasis in source], no matter how skilled, or how high standing in the community, has the right to act as though they are the **owner** [originally emphasized in italics] of a particular page. · · · Some contributors feel possessive about material they have contributed to Wikipedia. · · ·. Believing that an article has an owner of this sort is a common mistake people make on Wikipedia*"

How are socio-technical governance mechanisms used to mitigate the effects of ownership and territoriality? Large-scale social production systems, such as Wikipedia, require gover-nance mechanisms in order to direct the integration of dispersed knowledge resources in the process of value creation. Without such mechanisms, coordinating the work of assessing, selecting, shaping, integrating and oftentimes rejecting contributors' postings would be impos-sible [13]. Wikipedia employs a community-based governance model that is based on egalitar-ian principles, rather than on formal contracts and hierarchies [56]. Over the years, Wikipedia has developed an extensive set of mechanisms to ensure that the co-produced content is of high quality, including norms, policies, and technical tools for facilitating collaboration, resolving conflicts and fighting vandalism [14, 15, 29]. What is unique to Wikipedia gover-nance model is that it is collective and dynamic. That is, the community organically develops policies and procedures, and these continuously undergo changes by community members

[13]. Thus, governance is collective in the sense that it does not depend on any particular individual or a group of individuals within the system.

Central to Wikipedia's governance is the NPOV principle [25, 30–33], which promotes objectivity and acts to ensure that no one particular viewpoint dominates an article. NPOV is one of the five fundamental principles (or pillars) of Wikipedia. The principles (see https://en.wikipedia.org/wiki/Wikipedia:Five_pillars retrieved February 8, 2021) state that:

> "*We strive for articles that document and explain major points of view, giving due weight with respect to their prominence in an impartial tone. We avoid advocacy and we characterize information and issues rather than debate them. In some areas there may be just one well-recognized point of view; in others, we describe multiple points of view, presenting each accurately and in context · · · Editors' personal experiences, interpretations, or opinions do not belong.*"

Contributors to Wikipedia are requested to edit articles with a neutral perspective, representing views fairly, and trying to avoid bias. Wikipedia founder, Jimmy Wales, commented in a mailing list posting that "If a viewpoint is held by an extremely small (or vastly limited) minority, it does not belong in Wikipedia regardless of whether it is true or not and regardless of whether you can prove it or not." (https://lists.wikimedia.org/pipermail/wikien-l/2003-September/006715.html; retrieved February 8, 2021). However, biases, the opposite of NPOV, may arise when subjective information is too difficult to verify, when an issue is too complex to be fully represented (whereby generating a consensus requires considerable effort and expertise), or in cases when contributors intentionally try to impose their own ideology in an attempt to influence readers' viewpoints [25, 33, 56]. Enforcing NPOV has become a central concern for Wikipedians: the focus of many of the discussions found on articles' coordination spaces (i.e. TalkPages) [8] and dedicated Internet Relay Chat (IRC) channels [25] is on whether particular section of an article reflects the NPOV principle. Editorial activity is often not distributed evenly within a Wikipedia article, such that a few highly active contributors are responsible for a large share of the activity. Thus, in enforcing neutrality, the community tries to ensure that the "ownership" of an article's content is not concentrated in the hands of few contributors.

Governance is primarily intended to regulate counter-productive behavior. We suggest two potential mechanisms by which NPOV enforcement could affect the competitive dynamics. On the one hand, as intended, the efforts to ensure neutrality may serve to curtail excessive one-sided ownership of an article, and consequently directly impact contributors' attempts to own article fractions. On the other hand, the adherence to norms, policies and procedures could deter participation in co-production, leading to lower participation levels, which, in turn, might influence the quality of the co-produced article.

## 3 Research question

Notwithstanding the value of prior research, the scholarly literature on peer production has not investigated the interplay between individuals' cooperative/competitive dynamics and the socio-technical governance mechanisms. Our goal in this paper is, thus, to investigate Wikipedians' cooperative content production and the competition over content ownership, as well as to explore the way in which this competitive dynamic is shaped by the governance community's effort to ensure an NPOV. We recall that contributors derive benefits from both the collective production of a high-quality article (more applicable to those with a cooperative orientation) and from the ownership of content (more applicable to competitively-oriented

contributors). Contributors also incur costs in producing content (effort is a function of one's activity profile: creator or curator), coordinating and organizing work (largely applicable to cooperative contributors) and in complying with Wikipedia's neutrality-enforcement mechanisms (primarily for competitive contributors). Hence, we pose the following two research questions, which we investigate to answer in this research.

RQ1: What is the optimal strategy for a contributor who is attempting to balance the costs and benefits of peer production? Namely, how many sentences within an article should be owned by a contributor that is characterized by a certain cooperative/competitive orientation and a particular creator/curator activity profile?

RQ2: How does the community governance mechanism, particularly the attempt to ensure a NPOV, affect the dynamics underlying Wikipedia's co-production process?

## 4 A game theoretic model of Wikipedia's collaborative production and governance

A game-theoretic model should capture essential features of the modeled system. Below we present our model in a descriptive manner, and then introduce the formal notations in the following subsections. Socio-technical systems such as Wikipedia exhibit both cooperative and competitive dynamics. Our game-theoretic model aims to represent (*a*) contributors' cooperation in the production of high-quality articles as well as their competition to "own" articles" contents and (*b*) the community's attempt to ensure that articles present a neutral and balanced perspective. In other words, our model assumes contributors may be driven by two goals: the first is to contribute towards the creation of comprehensive encyclopedic entries and the second goal is to maximize ownership of article sections, such that content "owned" by that contributor survives multiple rounds of revisions of the article. Our model also assumes that the objective of Wikipedia's governance mechanisms enacted by the community is to ensure the neutrality of content, such that no single contributor takes ownership of large portions of an article. Wikipedia's governance has the broader goal of regulating behavior, such that the community thrives and the peer-produced content is of high quality. Our focus here is on one important dimension of governance: that of combatting the attempts to "color" the article according to one particular viewpoint, that is, to ensure objectivity and the lack of bias in Wikipedia's content [8]. Beyond the importance of this particular dimension of Wikipedia's governance (as evident by the NPOV pillar), objectivity is particularly relevant for our conceptualization, as it is directly impacted by contributors' effort to "own" content.

We model the interactions between the governance mechanisms and contributors' co-production as a Stackelberg (leader-follower) game [20, 21], where the community is the leader that determines a level of governance that increases the neutrality of an article. Given a set level of governance, contributors' interactions are modeled as a non-cooperative game, where each contributor's objective is to maximize both the article's comprehensiveness and the contributor's content ownership. A high level schematic of the interactive model between the community's goal of ensuring neutrality and contributors' cooperation/competition is presented in Fig 1.

Peer production on Wikipedia is a complex socio-technical process and contributions occur asynchronously over time. Our strategy for capturing these dynamics is temporal bracketing: recording a series of yearly "snapshots" of the process over a finite time horizon [71]. We use a static game-theoretic model to study these interactions, and test the model against empirical data for each of the yearly snapshots. Please note that modeling such a complex process requires some simplifying assumptions. We assume that contributors' activity profiles, namely, their position on the *creator-curator* continuum and their cooperative/competitive

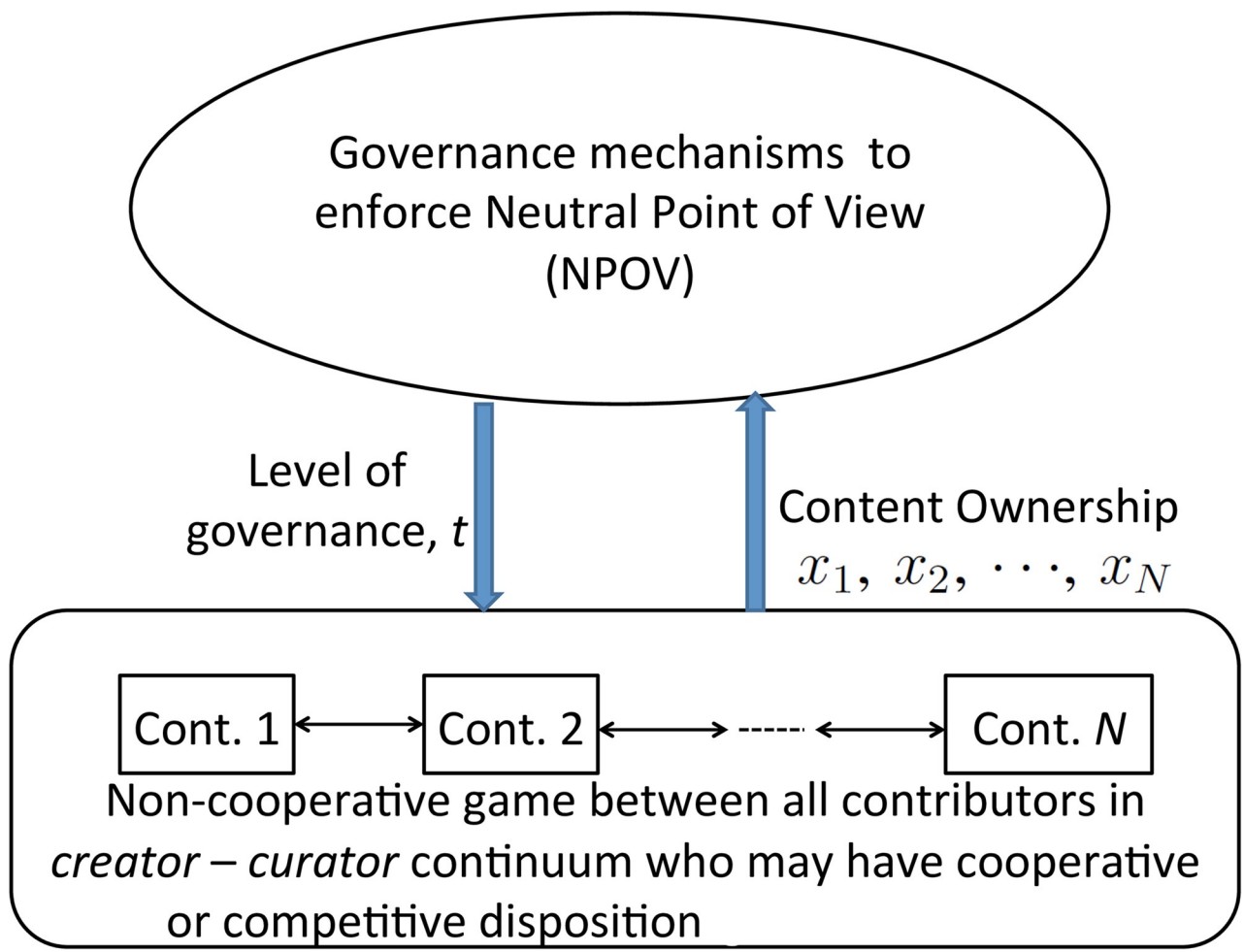

**Fig 1. A high level schematic of the interactive model between Wikipedia's governance to ensure neutral point of view (NPOV) and the interactions between contributors.**

orientation, are stable over time (at least within a yearly snapshot). This assumption is in line with prior research that demonstrated that the changes in contributors' activity profiles are infrequent [16]. We also assume that Wikipedia's governance structure, and in particular the goals of ensuring neutrality, are stable over time (as changes in policies are relatively infrequent). It should be stressed that while our model provides an approximation to the complexities of peer production, we also validated it using empirical data that captures Wikipedia's actual collaborative-authoring dynamics.

We model the net utility as the difference between the utility (the contributor's fractional ownership and the benefit derived from the co-production of high-quality articles) and the cost (a measure of all the efforts expended by the contributor). The optimal strategies are determined through the Nash equilibrium of the non-cooperative game that models the interactions between contributors. We analyze the conditions for the existence and uniqueness of a Nash equilibrium for the game and discuss its implications.

As noted, we model the interactions between the governance mechanism—namely, enforcing a neutral point of view- and contributors' co-production as a Stackelberg (leader-follower) game. The role of the neutrality-enforcing mechanism is to choose the optimal level of

**Table 1. List of notations used to set up the game theoretic model.**

| Notation | Description |
|---|---|
| $N$ | Number of contributors for a focal Wikipedia article |
| $x_i$ | Amount of contribution by the $i^{th}$ contributor to the focal article |
| $u_i$ | Utility of the $i^{th}$ contributor in the focal article |
| $c_i$ | Fractional (content) ownership of the $i^{th}$ contributor in the focal article |
| $f_i$ | Fixed cost incurred by the $i^{th}$ contributor (by participation in peer production and governance work) |
| $t$ | Level of neutrality enforcement |
| $\beta_i$ | Average size of the $i^{th}$ contributor's edits (across all articles he contributed to reflecting the contributor's activity profile: whether a creator (making larger edits) or a curator (making smaller edits). $\beta_i$ is also used as a proxy for the amount of effort exerted in making an edit |
| $w_i$ | A contributor's cooperative/competitive orientation, 0 indicating an entirely cooperative orientation and 1 indicating a competitive orientation. |
| $H(t)$ | Entropy of an article with governance level, $t$ |

enforcement, $t$, defined in our model in terms of the effort involved in complying with governance (i.e., when editing a sentence). The response of the $N$ contributors in the particular article is to make unilateral contributions that maximize their net utility (the difference between the benefit obtained from their content ownership as well as co-producing a comprehensive article and the efforts expended to make edits, maintain the quality of the article and comply with governance). A contributor may choose to make any number of contributions (i.e., sentences owned in an article). Our objective is then to determine for each contributor, the optimal levels of activity, so that each contributor's net utility is maximized. Please refer to Tables 1 and 2 for the list of notations/variables that are used in our analysis.

## 4.1 Modeling co-production: Contributors' cooperation and competition

We model the interactions of the $N$ content contributors to a particular Wikipedia article as a non-cooperative game. Please note that while our model assumes a given number of contributors, it does allow for contributors to make a choice regarding their participation. Namely, by deciding to own zero content contributors are actually deciding not to join the game. All contributors can determine their individual activity levels, namely the number of sentences the contributor would attempt to own, a choice that affects that contributor's utility and costs. The $N$ contributors in the particular article can make unilateral contributions − $x_1, x_2, \cdots, x_N$, respectively, where $x_i$ represents the $i^{th}$ contributor's overall activity (i.e., the number of

**Table 2. List of notations used to evaluate the Nash equilibrium of the game.**

| | |
|---|---|
| $x_i^*$ | Optimal amount of contribution by the $i^{th}$ contributor in the focal article which maximizes $u_i$ with respect to $x_i$ |
| $c_i^*$ | Fractional content ownership of the $i^{th}$ contributor within the focal article when she makes the optimal contribution, $x_i^*$ |
| $\alpha_i$ | A simplified notation to represent $\frac{1}{t+1+\frac{t\beta_i-1}{w_i}}$ |
| $\mathbf{D}$ | An $N \times N$ diagonal matrix with $N, 0, \cdots, 0$ along the diagonal |
| $\mathbf{D}_\alpha$ | A diagonal matrix with $\alpha_1, \alpha_2, \cdots, \alpha_N$ along the diagonal |
| $\mathbf{1}$ | A column vector in which all entries are 1, whose transpose is $\mathbf{1}^T$ |
| $\mathbf{P}$ | A matrix whose columns are orthonormal eigen vectors of $\mathbf{11}^T$ |
| $E\left(\frac{\beta}{w}\right) = \frac{1}{N}\sum_{i=1}^{N}\frac{\beta_i}{w_i}$ | Average ratio of size of edits and competitive orientation for all those contributing to the focal article |

sentences the contributor would like to own). That decision affects the comprehensiveness of the article and the content ownership yield. The strategy set, therefore, is the content added to the article by each contributor $x_1, x_2, \cdots, x_N$. Our model assumes both self-interest (i.e., competitive) and cooperative motivation for contributors. We define a "weight", $w_i$ ($0 < w_i < 1$) to represent the extent to which one is competitive, where $w_i$ represents the cooperative aspect. When $w_i \to 0$, it indicates that the $i^{th}$ contributor is cooperatively oriented and the $i^{th}$ contributor is considered to be competitively oriented if $w_i \to 1$. Additionally, we define a "weight," $\beta_i$ ($0 \leq \beta_i \leq 1$; i.e., $\beta_i$ is *continuous* and can take all possible values in the interval, [0, 1]) that identifies a contributor as either a creator or a curator (one may be involved in a combination of creation and curation tasks). Creators are characterized by large-size edits (i.e., larger $\beta_i$'s); curators' edits are smaller (i.e., smaller $\beta_i$'s).

In terms of their benefits, cooperatively, a contributor derives value from the creation of a high-quality and comprehensive article. Particularly, comprehensiveness is considered a key indicator of information quality [45] For example the number of words or sentences is a very good predictor of article quality [46, 47]. We model this aspect of the contributor's utility as the sum of the article portions that are "owned" by all those contributing to the focal article, $\sum_{i=1}^{N} x_i$ (total number of sentences in the article).

Competitively, the goal of individual contributors is to advance their personal point of view and "own" contents of an article. We model the competitive component of utility for each contributor in the non-cooperative game as the contributor's fractional ownership in the article. The actions of contributors are either originating new content or changing existing content to the co-created artifact. Contributors may come to "own" portions of the article (in our case, in terms of sentences), whereby each sentence a contributor comes to "own" indirectly reduces others' fractional ownership. The $i^{th}$ contributor has fractional content ownership,

$$c_i = \frac{x_i}{\sum_{j=1}^{N} x_j}. \tag{1}$$

Each contributor incurs a cost in making the contribution, modeled as the sum total of:

a.  the effort expended in producing content and editing an article. This cost element depends on the average effort of taking ownership of a sentence as well as the number of sentences the contributor eventually likes to own, i.e., $x_i$. We assume that changing an existing sentence requires less effort than originating a new sentence, such that the effort expended by a contributor is a linear function the contributor's position in the creator-curator continuum; ($\beta_i$), in line with prior works in the area of software development which approximated coding effort based on the quantity of code that is produced (i.e., lines of code) [39, 40], we assume that effort is a linear function of edit size (i.e., one's position in the creator-curator continuum; $\beta_i$). Thus, the effort involved in editing a single sentence is a function of the contributor's activity profile, $L\beta_i$, where $L$ is a constant. Hence, the overall effort contributor $i$ makes in editing the focal article is $L\beta_i x_i$. Please note that this cost is specific to each contributor and is independent of the contributor's cooperative/competitive orientation.

b.  the effort associated with participation in coordination and administrative work, such as editing `the articles' TalkPages`. Such activities take place on auxiliary pages and are independent of one's content production activity. This cost element is more applicable for the cooperative contributors, i.e., $f_i(1 - w_i)$.

c.  the effort of complying with Wikipedia's rules and policies, in particular, neutrality-enforcing policies. This effort is intended to regulate only the self-interest activities (i.e., attempting to "own" article portions) and thus is more applicable to competitive contributors. This

cost element depends on the governance-compliance effort associated with editing a single sentence, (similar for all contributors) as well as on the number of sentences edited by a contributor. Therefore, for the $i^{th}$ contributor with competitive orientation, owning $x_i$ amount of sentences requires expending an effort $tw_ix_i$ to comply with the level of neutrality enforcement, $t$.

In sum, the net utility experienced by the $i^{th}$ contributor, $u_i$, can be written as the difference between utility of contributor $i$ and the total effort expended by contributor $i$. Formally stated:

$$u_i = w_ic_i + (1 - w_i)\sum_{i=1}^{N}x_i - L\beta_ix_i - tw_ix_i - f_i(1 - w_i)$$

$$= \frac{w_ix_i}{\sum_{j=1}^{N}x_j} + (1 - w_i)\sum_{i=1}^{N}x_i - (L\beta_i + tw_i)x_i - f_i(1 - w_i). \tag{2}$$

Each contributor then determines the optimal amount of sentences owned, (i.e., $x_i^*$), that maximizes his or her net utility, $u_i$ *when the contributions made by all other contributors, $x_j$, $j \neq i$ are fixed.* It is therefore observed that the net utility obtained by the $i^{th}$ contributor not only depends on the strategy of the $i^{th}$ contributor (i.e., $x_i$), but also on the strategies of all the other contributors (i.e., $x_j$, $j \neq i$). Therefore, $u_i$ is also sometimes denoted as $u_i(x_i, \mathbf{x}_{-i})$ where $\mathbf{x}_{-i}$ represents the vector, $\begin{bmatrix} x_1 & x_2 & \cdots & x_{i-1} & x_{i+1} & \cdots & x_N \end{bmatrix}^T$, i.e., all elements of the vector, $\mathbf{x} = \begin{bmatrix} x_1 & x_2 & \cdots & x_N \end{bmatrix}^T$ except the $i^{th}$ element. This results in a non-cooperative game of complete information [72] between the contributors. The optimal $x_i$, $\forall i$ (denoted as $x_i^*$), which is determined by maximizing $u_i$ in Eq (2), when all other $x_j$'s, $j \neq i$ are fixed, i.e., the vector, $\mathbf{x}_{-i}$ is fixed, is then the Nash equilibrium of the non-cooperative game where no contributor can make a unilateral change.

The unique Nash equilibrium, $\mathbf{x}^* = \begin{bmatrix} x_1^* & x_2^* & \cdots & x_N^* \end{bmatrix}^T$, can be obtained in closed-form as follows. $\mathbf{x}^*$ is obtained by determining the best response for contributor $i$, $x_i^*$ which maximizes its net utility, $u_i$ (given by Eq (2)) when the contributions made by all other contributors, i.e., $x_j$, $j \neq i$ are known. However, it is noted that *every contributor attempts to do the same.* In other words, the unique Nash equilibrium, $\mathbf{x}^* = \begin{bmatrix} x_1^* & x_2^* & \cdots & x_N^* \end{bmatrix}^T$, is the vector that maximizes the net utility, $u_i$, in Eq (2), $\forall \underline{i}$. Applying the first order necessary condition to Eq (2), $x_i^*$ is obtained as the solution to

$$\left.\frac{\partial u_i}{\partial x_i}\right|_{x_i = x_i^*} = \frac{\sum_{k=1, k\neq i}^{N}w_ix_k^*}{\left(\sum_{j=1}^{N}x_j^*\right)^2} + (1 - w_i) - (L\beta_i + w_it) = 0, \quad \forall i \tag{3}$$

subject to the constraints $x_i^* \geq 0$, $\forall i$.

From Eq (3), we obtain $\frac{\partial^2 u_i}{\partial x_i^2} = -\frac{2\sum_{k=1, k\neq i}^{N}w_ix_k}{\left(\sum_{j=1}^{N}x_j\right)^3} < 0$, $\forall i$, when $x_i \geq 0$. Thus, $u_i$ is a concave function of $x_i$ and $x_i^*$, which solves Eq (3) subject to $x_i^* \geq 0$, $\forall i$, is a local as well as a global maximum point. In other words, *the non-cooperative game has a unique Nash equilibrium,* The vector of contributions, $\mathbf{x}^* = \begin{bmatrix} x_1^* & x_2^* & \cdots & x_N^* \end{bmatrix}^T$, can be obtained by numerically solving the system of $N$ non-linear equations specified by Eq (3). However, to study the effect of the contributors' type (i.e. *creators* or *curators* based on their $\beta_i$'s) and the level of neutrality enforcement, $t$, on contributors' strategies, it is desirable to obtain an expression that relates the vectors, $\mathbf{x}^*$, $\mathbf{x} = [x_i]_{1 \leq i \leq N}$, $\boldsymbol{\beta} = [\beta_i]_{1 \leq i \leq N}$ and $t$.

Solving Eq (3), we propose the following theorem that provides the unique Nash equilibrium of the non-cooperative game.

**Theorem 1:** The unique Nash equilibrium of the non-cooperative game can be obtained as

$$x_i^* = \frac{\sum_{j=1}^N \left(\frac{L\beta_j - 1}{w_j}\right) - (N-1)\left(\frac{L\beta_i - 1}{w_i}\right) + t + 1}{\left(\sum_{j=1}^N \left[\left(\frac{L\beta_j - 1}{w_j}\right) + t + 1\right]\right)^2}. \tag{4}$$

*Proof:* See Appendix-1 in the S1 File.

**Theorem 2:** The unique Nash equilibrium $\mathbf{x}^*$, is feasible, *i.e.*, $x_i^* > 0, \forall i$ if and only if

$$(N-1)\left(\frac{L\beta_i - 1}{w_1}\right) < t + 1 + \sum_{j=1}^N \left(\frac{L\beta_j - t}{w_j}\right)$$

$$\text{or } (N-2)\left(\frac{L\beta_i - 1}{w_1}\right) < t + 1 + \sum_{\substack{j=1 \\ j \neq i}}^N \left(\frac{L\beta_j - 1}{w_j}\right). \tag{5}$$

*Proof:* The proof follows by just setting $x_i^*$ in Eq (4) to be $x_i^* \geq 0$.

Intuitively, Eq (5) implies the following. Let all the other contributors as well as the neutrality enforcement represent "adversaries" of contributor $i$. The right hand side shows the total adversarial activity level against contributor $i$. The left hand side shows a term similar to the total activity level in the favor of contributor $i$ if contributor $i$ were to duplicate herself against each individual adversary. Eq (5) then says that a contributor makes a non-zero contribution to an article only if the contributor has to expend effort in his/her favor, that is less than the effort or activity level of the adversaries.

**Theorem 3:** The fractional content ownership of contributor $i$ at the Nash equilibrium, $c_I^*$, is

$$c_i^* = \left(\frac{x_i^*}{\sum_{j=1}^N x_j^*}\right)^+ = \left[1 - \left(\frac{(N-1)\left(\frac{L\beta_i - 1}{w_i}\right) + t + 1)}{\sum_{j=1}^N \left(\frac{L\beta_j - 1}{w_j} + t + 1\right)}\right)\right]^+, \tag{6}$$

where for any, $\theta$, $\theta^+ = \max(\theta, 0)$ and is non-zero only if Eq (5) is satisfied, i.e., if the Nash equilibrium is feasible.

*Proof:* The expression follows by substituting $x_i^*$ obtained from Eq (4) into Eq (1) and setting $c_i^* \geq 0$.

**Theorem 4:** Asymptotically, (i.e., when the number of contributors, $N$, becomes very large), $c_i^* > 0$ for only those contributors for whom $\frac{\beta_i}{w_j} < E\left[\frac{\beta}{\mathbf{w}}\right]$, where $E\left[\frac{\beta}{\mathbf{w}}\right] \triangleq \frac{1}{N}\sum_{j=1}^N \frac{\beta_j}{w_j}$, is the *average size of edits divided by the competitiveness weightage* of all contributors making changes to the focal Wikipedia article. In other words, when the number of contributors in an article is significant, only those contributors whose edit size (correlated with their position on the creator-curator continuum) per their competitiveness weight is less than the group's average end up with non-zero ownership.

*Proof:* Eq (6) can be re-written as

$$
c_i^* = \left[ 1 - \left( \frac{\dfrac{L\beta_i - 1}{w_i} + t + 1}{\dfrac{1}{N-1} \sum_{j=1}^{N} \dfrac{L\beta_j - 1}{w_j} + \dfrac{N}{N-1}(t+1)} \right) \right]^+
$$

$$
= \left[ 1 - \left( \frac{\dfrac{L\beta_i - 1}{w_i} + t + 1}{\dfrac{N}{N-1} \dfrac{1}{N} \sum_{j=1}^{N} \dfrac{L\beta_j - 1}{w_j} + \dfrac{N}{N-1}(t+1)} \right) \right]^+ .
$$

(7)

Asymptotically, i.e., as the number of contributors, $N$, becomes large $c_i^*$ is evaluated by taking $\lim_{N \to \infty}$ in Eq (7), to obtain

$$
c_i^* = \left[ 1 - \left( \frac{L\left(\dfrac{\beta_i}{w_i}\right) + t + 1}{LE\left[\dfrac{\boldsymbol{\beta}}{\mathbf{w}}\right] + t + 1} \right) \right]^+
$$

$$
= \left( \frac{LE\left[\dfrac{\boldsymbol{\beta}}{\mathbf{w}}\right] - L\left(\dfrac{\beta_i}{w_i}\right)}{LE\left[\dfrac{\boldsymbol{\beta}}{\mathbf{w}}\right] + t + 1} \right)^+ ,
$$

(8)

which is non-zero if and only if $\frac{\beta_i}{w_i} < E\left[\frac{\boldsymbol{\beta}}{\mathbf{w}}\right]$.

Note that the optimal $x_i^*$ in Eq (4) and the optimal fractional ownership, $c_i^*$ in Eq (6), do not depend on the fixed cost, $f_i$. This is because, the fixed cost does not change with changing $x_i$ (from Eq (2)) and hence has no bearing on the optimal $x_i^*$. Similarly, from Eq (1), the fractional ownership, $c_i$, is independent of $f_i$ and therefore, $c_i^*$ in Eq (6) is also independent of $f_i$. The condition in Eq (5) and the expression in Eq (6) have the following interesting implications.

- From Eq (6), the ownership of contributors depend on the $\beta_j$ of *all the contributors*. This is intuitively correct in a peer production project such as Wikipedia because changes to the produced artifact are made by multiple contributors and the ownership held by a contributor will depend on the activity of all the contributors co-producing the artifact.

- In Eq (4), the level of neutrality enforcement, $t$, appears as a linear factor in the numerator and as a second order quadratic factor (power of 2) in the denominator. Therefore, when neutrality enforcement, $t$ increases, the overall content owned by a contributor decreases. This implies that when the "tax" imposed on contributors in terms of complying with NPOV norms, policies and procedures is too high it outweighs the benefits associated with content ownership along with with making an article comprehensive. Thus, contributors stop co-production work.

- The $c_i^*$ given by Eqs (6) and (7) are the exact expressions for the content ownership. The asymptotic behavior in Eq (8) is used only to reach the conclusion that when the number contributors in an article is significant, only those contributors whose $\frac{\beta_i}{w_i}$ is below the group's average, survive the peer production process and end up owning non-zero percentage of content in the article. For all the numerical analysis, only Eq (6) is used.

A low value of $\frac{\beta_i}{w_i}$, indicates low $\beta_i$ (i.e., a contributor making small edits acting as a *curator*) and having $w_i$ closer to 1 (i.e. having a competitive orientation). In other words, if a contributor is entirely cooperative, then $w_i \to 0$ and $\frac{\beta_i}{w_i} \to \infty$ and becomes more than the average, $E\left[\frac{\boldsymbol{\beta}}{\mathbf{w}}\right]$. Thus, *contributors who are purely cooperative do not own any content*. This is a straightforward result, since cooperative contributors are primarily interested in producing high-quality articles, and less driven by the desire to "own" articles.

Competitive participants ($w_i \to 1$), with large edit sizes ($\beta_i$'s), who are creators of content (i.e., with substantial changes per sentence edited) also do not own content since $\frac{\beta_i}{w_i}$ becomes large. In other words, *only those with a competitive orientation who choose to act as curators making small edits end up owning significant portion of the content.* One might expect that the creators who contribute more content (and in the process exert more effort) would end up owning much of an article's contents. In contrast, the results of our game-theoretic analysis implies that when competing over content ownership in the presence of Wikipedia's governance to ensure neutrality, and when controlling for one's cooperative/competitive-orientation, the creators of content who make on average large contributions would eventually not own any content.

## 4.2 Neutrality enforcement and its effect on co-production

The community governance factor in our model is the result of contributors' aggregate governance effort. The level of governance, $t$, should be chosen so as to maximize the effectiveness of peer production. Our model assumes that the main objective of Wikipedia's governance is to reduce bias and ensure that all points of view are reflected in an article. Assuming a sufficiently large sample, the set of contributors working on an article would sample a range of viewpoints. The more one contributor (or a small subset of the contributors) is able to dominate the discussion (i.e. take ownership of large portions of the article), the more likely the article is to reflect a single side. In contrast, when content ownership within an article is more equally distributed and when multiple contributors own sections of the article, a more NPOV is expected. Thus, our notion of article neutrality reflects equal ownership of an article's contents. We are less interested here in the particular biases that may be present in the content added by each contributor.

Building on information theory [73], we use the metric of entropy to estimate the amount of bias in the distribution of content ownership. We note that the entropy metric was employed in prior studies in organization science [74], and in particular in estimating the distribution of work in Wikipedia [8]. In information theoretical terms, if a source emits $N$ possible symbols, $\mathbf{s} = \begin{bmatrix} W_1 & W_2 & W_3 & \cdots & W_N \end{bmatrix}^T$ with probability distribution, $p_i$, i.e., the source emits symbol, $W_i$, with probability, $p_i \geq 0$, $\sum_{i=1}^N p_i = 1$, then the entropy of the source, $H$ is defined as [75]

$$H = \sum_{i=1}^N p_i \ln \frac{1}{p_i} = -\sum_{i=1}^N p_i \ln p_i. \tag{9}$$

In any Wikipedia article, the fractional ownership of all contributors, $c_i^*$ are all non-negative and satisfy the condition, $\sum_{i=1}^N c_i^* = 1$. Therefore, a page can be viewed as as an information source with the $N$ contributors playing the role of the $N$ possible symbols and the fractional content ownership of each contributor, $c_i^*$, playing the role of $p_i$. The entropy of the page can

then be written as

$$H(t) = \sum_{i=1}^{N} c_i^* \ln \frac{1}{c_i^*} = -\sum_{i=1}^{N} c_i^* \ln c_i^*. \tag{10}$$

In Eq (10), the entropy, $H(t)$, may depend on other factors, e.g., the optimal contributions, $x_i^*$ $1 \leq i \leq N$, obtained from Eq (4), which in turn, depend on $\boldsymbol{\beta}$ and $\mathbf{w} = \begin{bmatrix} w_1 & w_2 & w_3 & \cdots & w_N \end{bmatrix}^T$. However, we represent the entropy as $H(t)$ because the neutrality enforcement, $t$, is the only parameter in $H$ that represents the action taken by Wikipedia's governance mechanisms. Our model's optimization function for neutrality enforcement (maximizing entropy in contributors' ownership) is normalized (each article's entropy is calculated in relation to the maximum entropy of any article, which is 1).

It was shown in information theory [75] that the expression of entropy, $H(t)$, in Eq (10) is maximized when $c_i^* = \frac{1}{N}, \forall i$. i.e., when all contributors make equal contribution, which corresponds to the scenario when there is complete neutrality. Therefore, we use the entropy in Eq (10) as the objective function for determining the optimal level of neutrality enforcement for the Stackelberg game. Namely, the objective function for neutrality enforcement in our model is to maximize the entropy in content ownership within an article.

The optimal level of neutrality enforcement, $t$, is the value of $z$ that maximizes $H(z)$ in Eq (10), which is obtained by replacing $t$ with $z$ in Eq (6). Then,

$$t = \arg \max_z H(z), \text{ i.e., } \frac{\partial H}{\partial z}\bigg|_{z=t} = 0. \tag{11}$$

From Eqs (6), (10) and (11), the optimal value of $t$ that maximizes the entropy is the value of $t$ that makes $c_i^* = 0, \forall i$, i.e., $t \rightarrow \infty$. See Appendix-2 in the S2 File for details. Our analysis indicates that while neutrality enforcement levels remain moderate, our prior results from the non-cooperative game remain unchanged, such that those with a cooperative orientation who are content creators (i.e., make above-average-size contributions) end up not owning content. However, when neutrality enforcement levels exceed a certain threshold the non-cooperative model collapses, ownership becomes a direct function of neutrality enforcement (rather than contributors' efforts). Namely, from Eq (6), when $t >> NE(\boldsymbol{\beta})$ (i.e. when the amount of effort a contributor has to exert to overcome neutrality enforcement is significantly- typically one order of magnitude- larger than the aggregate effort that all contributors exert in making unit contributions to the article), $c_i^* \approx \frac{1}{t} \rightarrow 0$. In other words, ALL contributors end up owning an insignificant but equal portion of the article's contents. Therefore, the optimal value of $t$ is obtained by solving

$$t = \arg \max_z H(z), \text{ i.e., } \frac{\partial H}{\partial z}\bigg|_{z=t} = 0,$$
$$\text{subject to the constraint}$$
$$t \leq z^*, \tag{12}$$

where $z^*$ represents the maximum level of neutrality enforcement allowed in order to maintain the neutrality of an article above a required threshold. The resulting Stackelberg game model is depicted in Fig 2.

The complete model for the leader-follower game between the governance mechanism and the competitive self-interested contributors suggests that at the Stackelberg equilibrium, the governance mechanism operates in such a manner to result in an implicit outcome that keeps the entropy high, i.e., the difference between maximum and minimum fractional content

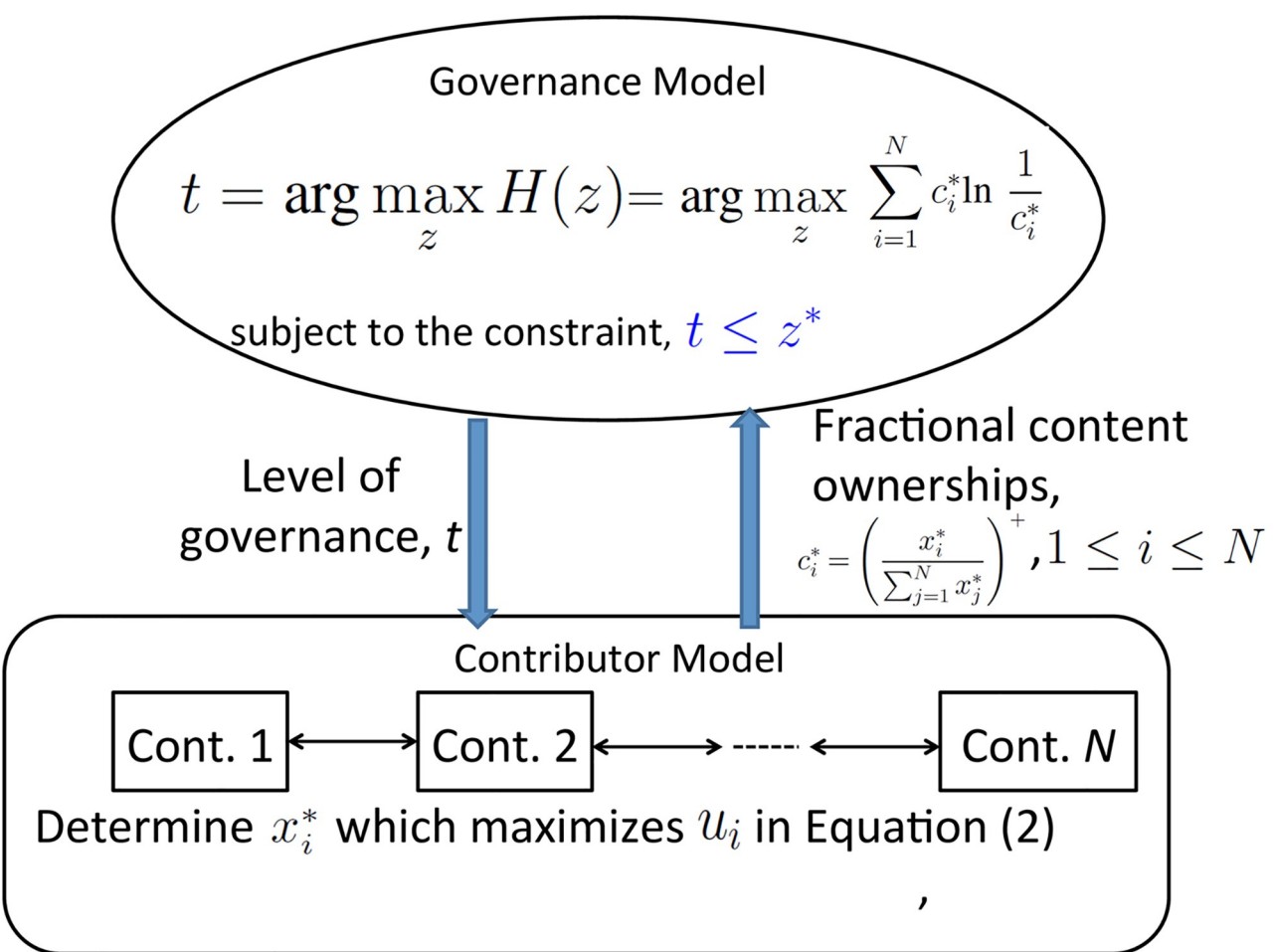

**Fig 2. The comprehensive Stackelberg game depicting the complete interaction between Wikipedia's governance to ensure neutrality and cooperation/competition peer production dynamics.**

ownership is kept low, thus ensuring the article's objectivity. Further, our model limits governance levels to an upper threshold.

## 5 Verification of the analysis with empirical data

To verify the game theoretic analysis, we compared its predictions against data from Wikipedia. In line with prior studies in the area [26], we employed a double-stratified sampling procedure, randomly selecting 1000 articles from the January 2012 dump of the English Wikipedia. Our strata were based on: (*I*) the number of revisions the article has gone through (i.e., article maturity) and (*II*) the article's topical domains. This sampling strategy is important given that collaboration patterns could differ across articles in different phases of their life cycle across topical domains. Given the power law distributions in the number of articles' revisions [76], we used the following four maturity strata: (*a*) 1–10 revisions; (*b*) 11–100; (*c*) 101–1000; and (*d*) more than 1000 revisions. The topical strata were based on Wikipedia's categorization system, using the main topics scheme. The English Wikipedia main topic categorization scheme is developed by the community and is subject to frequent changes; see http://en.wikipedia.org/wiki/Category:Main_topic_classifications. With 4 maturity strata and 25 topical categories, we have 100 cells with 10 randomly selected articles in each (i.e. 250 articles in each maturity

stratum and 40 articles in each topical category). Altogether, this sample contained: 1000 articles and 721,806 editing activities (i.e. article revisions), made by 222,119 contributors. In line with prior works in the area [8], we assume that for non-registered contributors, a unique IP address corresponds to a distinct contributor. About half of the contributors in our sample are non-registered. After excluding articles with a single contributor, we were left with 864 multi-contributor articles, which make up the sample of our analysis.

We began by counting the number of distinct contributors ($N$) and the total number of edits made by each contributor, across all articles and the total number of sentences edited by the $i^{th}$ contributor ($1 \leq i \leq N$), $\zeta_i$, across all of the articles in our sample. Next, we calculated the average amount of effort expended by the contributor when editing a sentence. A contributor's average edit size, $L\beta_i$, was calculated based on his activity profile across all articles within the sample to which he contributed. The contributor's position on the creator-curator continuum, $\beta_i$, was estimated based on the average size of edits per sentence, calculated using the Levenshtein distance [77]. The Levenshtein distance is a string metric used in information theory and computer science for measuring the difference between two sequences (or between two text segments) and is defined as the minimum number of single-character edits (i.e. insertions, deletions or substitutions) required to change one word into another [78]. This metric has been commonly used in the study of Wikipedia to estimate the scope of editing activities [79]. Based on this calculation, we were able to approximate the size of all edits made by the $i^{th}$ contributor ($1 \leq i \leq N$), $e_i$, to all articles in our sample. We then computed the average edit size of contributor $i$ per sentence, $L\beta_i$, $L\beta_i = \frac{e_i}{\zeta_i}$. Note that $\beta_i$ is a trait of the contributor, reflecting her activity profile (whether a *creator* or a *curator* of content) such that it remains the same across all articles that we analyzed.

In order to approximate a contributor's cooperative/competitive orientation, we relied on their functional roles within Wikipedia, as reflected by their access privileges. Wikipedia community has developed a comprehensive and detailed set of procedures for governing the collaborative editing process, including a well-defined scheme of access privileges [80]. Very broadly speaking, these roles could be organized in four strata, reflecting contributors' commitment and involvement within the community: unregistered members, registered members, privileged members (holding a special privilege), and core members (i.e. administrators) [81] Data regarding contributors' functional roles were accessed through Wikipedia's API. We relied on these strata in order to estimate the extent to which one is competitive (i.e. $w_i$) within the community: the greater one's rights and responsibilities within the Wikipedia community, the more one is considered to have a cooperative orientation. Specifically, we assigned the values of 0.1, 0.4, 0.6, and 0.9 to the different strata (e.g., an administrator is assigned $w_1 = 0.1$, reflecting the least competitive and most cooperative orientation).

We measured contributors' self-interest benefit (i.e. ownership of articles' contents) and costs (namely, the effort in making an edit) by calculating the fractional ownership for each contributor in each article of our sample using the algorithm in [38]. The content ownership algorithm tracks the evolution of content, recording the number of sentences owned by each contributor at each revision, until the study's end date. The algorithm uses a sentence as the unit of analysis, where each full sentence is initially owned by the contributor who first added it; as content on a wiki page evolves, a contributor loses a sentence when more than 50% of that sentence's content (i.e. the words in the original sentence) is deleted or revised by others (thus, a contributor making a major revision to an existing sentence can take ownership of that sentence). If no single contributor "owns" more than 50% of a sentence, that sentence becomes ownerless. The output of the algorithm indicates the number of sentences (and the fraction of sentences within the focal Wikipedia article) originated by each contributor that

persisted in the most recent version of the article. We defined $\hat{x}_i$ as the number of sentences owned by contributor $i$. A contributor's fractional ownership in a particular article is obtained as $\frac{\hat{x}_i}{\sum_{j=1}^{N} \hat{x}_j}$.

The number of contributors in the 864–article sample was found to follow a uniform distribution with the average number of contributors being 124.6 and a standard deviation, 71.9. We also found that contributors are more likely to make smaller edits per sentence than larger edits, with the distribution of $L\beta_i$ resembling an exponential probability density function. The average $L\beta_i$ was 6.8 (measured in Levenshtein distance) and the standard deviation was 6.2. The average content ownership in an article for users, $c_i^*$, was found to have a mean of 8% and a standard deviation of 0.55.

Employing the $\beta_i$'s and $w_i$'s thus obtained, we used the expression in Eq (6) to determine our model's prediction for a contributor's fractional ownership in a particular Wikipedia page. Eq (6) provides us with a contributor's expected fractional ownership. We repeated this analysis for all Wikipedia articles in our sample and for all contributors in each article. Our empirical analysis suggests that a contributor's creator/curator activity profile ($\beta_i$) and cooperative/competitive orientation ($w_i$) are two distinct variables, as the correlation between their estimators is quite low (Pearson correlation = 0.167).

We empirically tested our model's predictions regarding the relation between contributors' average edit size (i.e. a proxy for the effort typically exerted by a contributor when making an edit, reflecting their position on the *creator-curator* continuum), their cooperative/competitive orientation and their resulting content ownership. In trying to gain a better understanding of the temporal dynamics underlying the co-production competition over an article's content, we applied a temporal bracketing technique by breaking each article into yearly brackets (i.e. Year 1 since inception, Year 2, etc.). We then created subsets by yearly periods, capturing the states of all articles after their first year of operation, second year etc., and calculated contributors' average effort (i.e. their creator-curator profile), cooperative/competitive orientation and fractional ownership at the end of each year in an article's life. Next, for each of these yearly subsets, we repeated the analysis described above, comparing the model's prediction to actual fractional ownership. Our analytical results indicate that the number of sentences owned by a contributor, $x_i$, hinges on both $w_i$ and $\beta_i$. Naturally, it is expected that $\hat{x}_i$ correlates with these two variables. The correlation between $\hat{x}_i$ and $w_i$ is 0.678 (i.e. contributors that have a competitive orientation tend to own more sentences). The correlation between $\hat{x}_i$ and $\beta_i$ is −0.571 (i.e., curators tend to own more sentences).

Our findings show that the model's predictions regarding contributors' fractional ownership are almost identical to the empirical findings. Please see Fig 3 for the results for the entire study period. Looking at the temporal brackets, we observe that already at the end of the year of inception (Year 1), the process converges and our model's predictions closely match the empirical data (see Fig 4). Fig 5 shows that from Year 2 onward, the results are almost identical to the findings for the entire period. These results suggest that as articles mature and the co-production process stabilizes, our empirical results become more aligned with the model's prediction.

We measured the Pearson's correlation coefficient value between the fractional ownership obtained by the game theoretic analysis and that obtained from the measured data (2 years from inception, and onward), and found it to be 0.789, thus demonstrating that the data closely follow our model's predictions. The stability of our findings across various stages of an article's life cycle solidifies the validity of our model, demonstrating that despite simplifying assumptions and using a static model, we were able to capture the temporal dynamic underlying cooperation and competition in Wikipedia's co-production of articles. In order to further

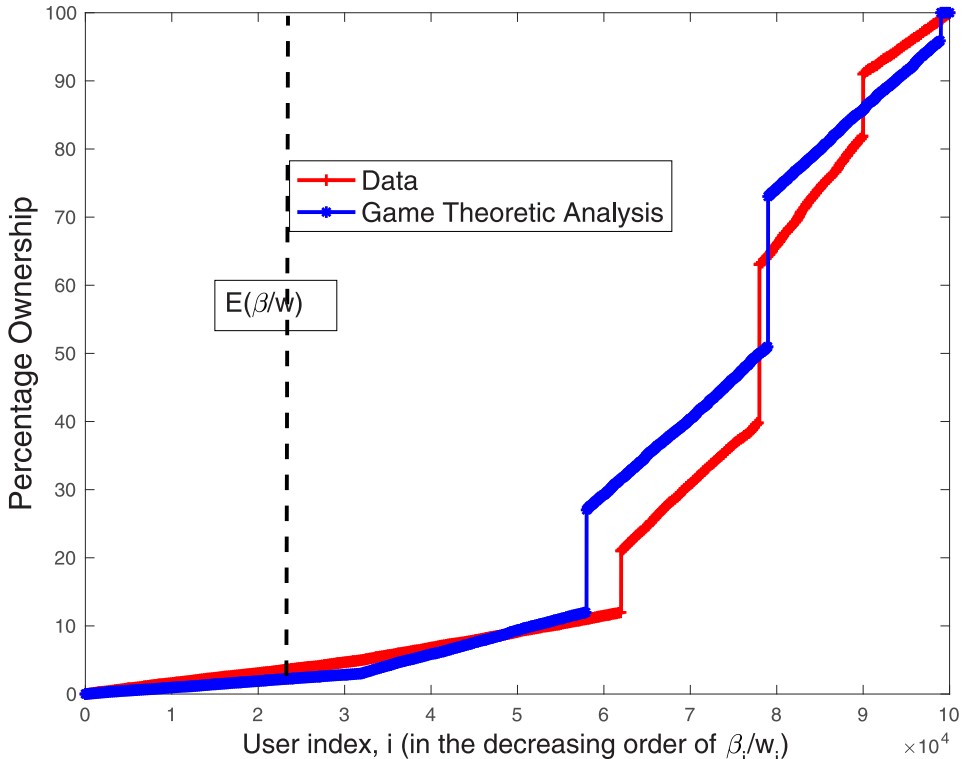

**Fig 3. The percentage ownership obtained from the data and that obtained from Eq (6) for 864 multi-contributor articles in our sample, *for the entire period of study*.** Results represent the aggregated solution for the 864 games (each game solved with the appropriate number of editors). Contributors are indexed according to the decreasing order of $\frac{\beta_i}{w_i}$'s.

test for the effect of articles' maturity, we analyzed the alignment between the model's prediction and the empirical findings by the number of contributors per article. We found that the alignment increases as the number of contributors per article grows (stabilizing at roughly 30 contributors per article). In order to verify the robustness of our model, we re-ran our analysis, this time excluding non-human editors (i.e. "bots"). Our findings indicate that the pattern of results for both the entire-period and the temporal bracketing analyses remains the same.

Trying to get a better sense of the (small) discrepancies in ownership values between the model's prediction and the empirical data, we explored whether the differences could be offset by establishing a linear fit function mapping. Let $\mathbf{a} \triangleq [\, a_1 \quad a_2 \quad a_3 \quad \cdots \quad a_N \,]$ (representing the ownership of the contributors obtained by the game theoretic analysis) and $\mathbf{d} \triangleq [\, d_1 \quad d_2 \quad d_3 \quad \cdots \quad d_N \,]$ (representing the ownership of the contributors obtained from the empirical data). For each Wikipedia article, we fit a function

$$\hat{d}_i = \rho a_i + \delta, \quad 1 \leq i \leq N, \tag{13}$$

where the parameters, $\rho$ and $\delta$, are obtained by the method of least squares [82]. We randomly selected 300 articles as training data in supervised learning to determine the linear regression coefficients, repeating the training/testing split 1000 times. The average error for the training data was 7.6%. We then ran a linear regression on the remaining 564 pages and tested the linear fit as well as its significance by measuring the $p$−value. The error was found to be between 18–23% and the linear regression was statistically significant at $p < 0.05$. The implication of

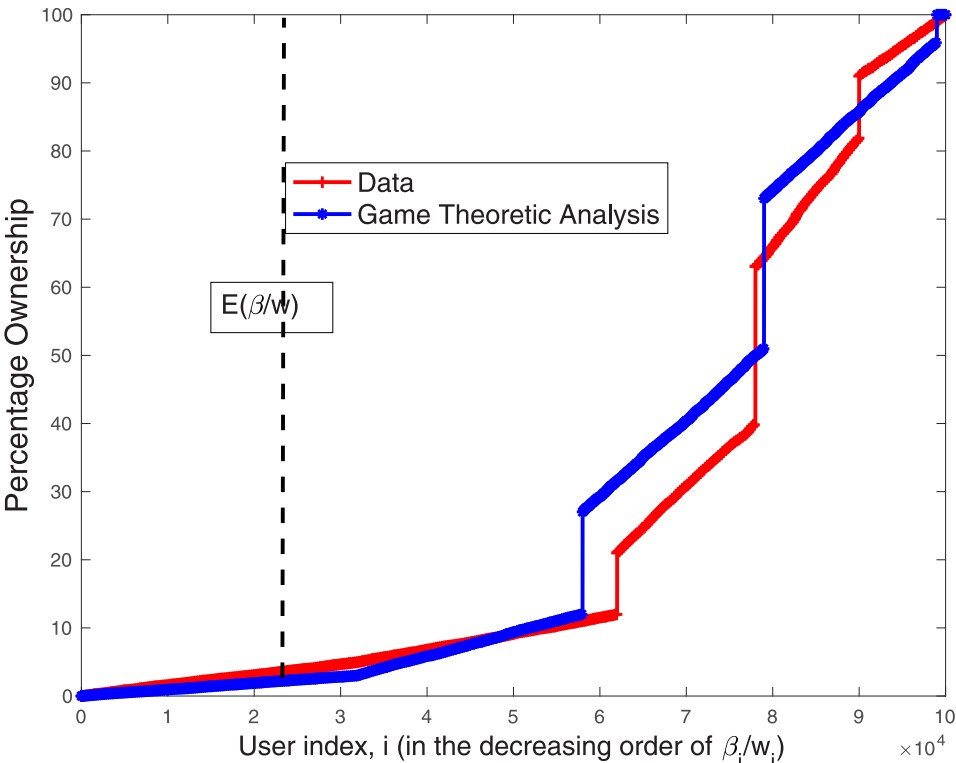

**Fig 4. The percentage ownership obtained from the data and that obtained from Eq (6) for 864 multi-contributor articles in our sample,** *for the first year of inception.* Results represent the aggregated solution for the 864 games (each game solved with the appropriate number of editors). Contributors are indexed according to the decreasing order of $\frac{\beta_i}{w_i}$'s.

this result is that the game theoretic analysis models the contributors' interactions in Wikipedia accurately up to a linear scaling factor.

We further employed a combination of the game theoretic analysis and findings from the empirical data to determine how governance levels influence the trade off between articles' neutrality and their comprehensiveness Fig 6 depicts the impact of governance level, *t*, on the neutrality of an article (measured by its entropy) and the comprehensiveness of the article (measured in terms of the article size). The entropy was measured from empirical data and the article size was obtained analytically using $\sum_{i=1}^{N} x_i^*$ in Eq (4). Fig 6 shows that reasonably low levels of governance $t \approx 6$ ensure neutrality (large amount of entropy $>0.9$), while maintaining an article's comprehensiveness (high value for $\sum_{i=1}^{N} x_i^*$). However, when the governance, *t*, becomes larger (closer to 100), the comprehensiveness of the article (article size) is reduced significantly yet the gain in neutrality (i.e. entropy) is marginal. This result suggests that that low levels of governance are sufficient for curbing much of the competition over content ownership, ensuring a relatively even distribution of ownership, while resulting in only a very minor reduction in activity levels, thus having a minimal impact on articles' comprehensiveness.

## 6 Discussion and conclusion

Whereas most research in the area has emphasized the cooperative aspects in Wikipedia's co-production, answering the call to shift the focus of online collaboration research to issues of

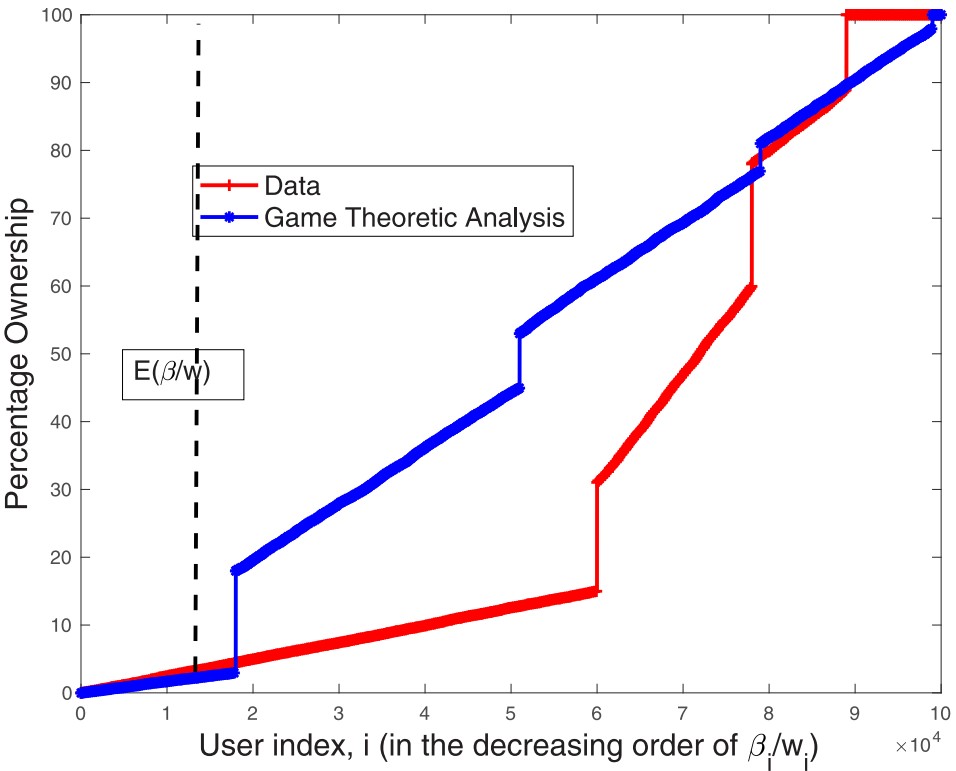

**Fig 5. The percentage ownership obtained from the data and that obtained from Eq (6) for 864 multi-contributor articles in our sample, _for the second year of inception._** Results represent the aggregated solution for the 864 games (each game solved with the appropriate number of editors). Contributors are indexed according to the decreasing order of $\frac{\beta_i}{w_i}$'s.

scarcity and competition [83], the focus of the current study is on the interplay between contributors' competition and cooperation. In particular, in this paper we developed a game-theoretic model of cooperation and competition in Wikipedia, under constraints of governance. We corroborated the model through an empirical study. We stress that we did not intend our game-theoretic model to be comprehensive; socio-technical systems such as Wikipedia are much too complex to be fully captured through a game-theoretic model. Instead, we attempted to model a few essential features of Wikipedia's co-production process, namely contributors' cooperation in producing high-quality and comprehensive articles, and their competition over content ownership, while taking into account the community's efforts to ensure that articles represent a balanced perspective.

The key results of our game-theoretic analysis are as follows:

- Analytically, we found that the contributor's characteristics, or more specifically, the ratio between the contributor's position on the creator-curator continuum and the contributor's cooperative/competitive orientation is the factor that determines who ends up owning content. When this ratio is smaller than the group's average, the contributor maintains ownership over portions of the article. Namely, under the governance mechanisms, the fractional content that is eventually owned by a contributor is higher for curators (i.e., with a typical small-size edit per sentence) with a competitive orientation (i.e., peripheral community members). In essence, creators with a cooperative orientation lose ownership of the article. This result was corroborated through an empirical analysis of 219, 811 distinct contributors

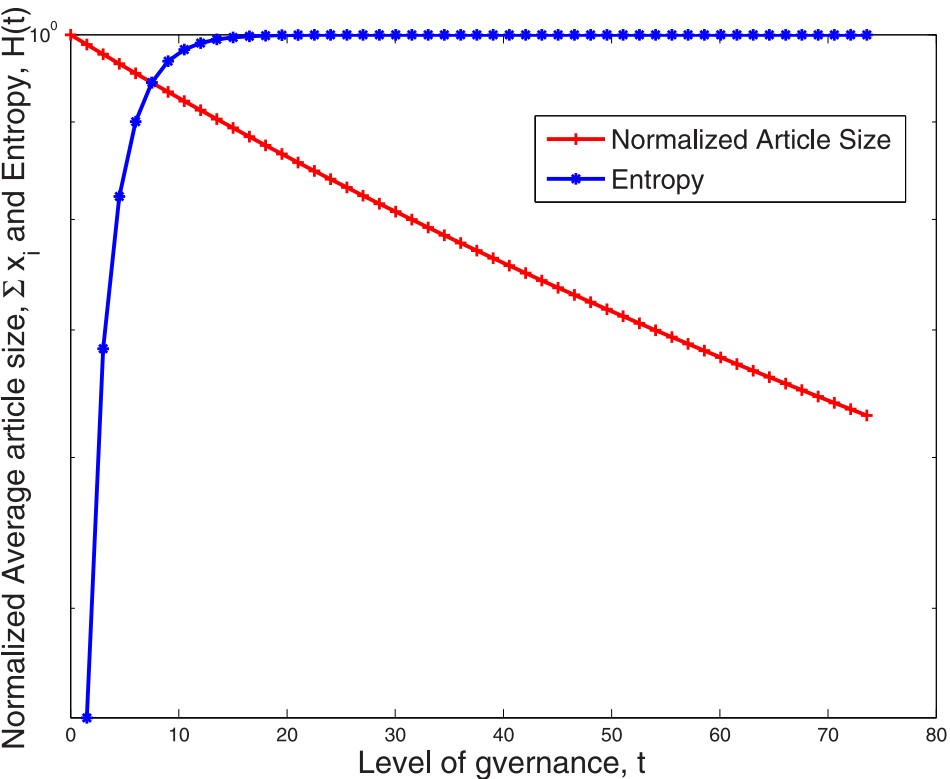

**Fig 6. The effect of governance level, *t*, on the article size, obtained analytically using $\sum_{i=1}^{N} x_i^*$ in Eq (4) and the entropy, *H(t)* obtained from empirical data.** For *H(t)* we just measured the entropy of each article using Eq (10) but with the values of $c_i^*$ from the data. The articles were indexed in the same order for which the normalized number of sentences were calculated.

co-producing 864 representative articles over an eleven-year period, which used the proxies mentioned earlier. Employing a temporal bracketing approach, we analyzed the data for each yearly bracket independently, to find that already from the second year of an article's life and onward the empirical pattern of results converges to our analytical predictions.

- Analytically, we show that excessive governance should be curtailed, by identifying and maintaining a permissible upper limit, beyond which it discourages contributors from making contributions to an article, bringing the co-production process to a halt. Furthermore, our empirical analysis suggests that a low level of governance i s optimal for ensuring neutrality while maintaining articles' comprehensiveness.

We note that prior studies have discussed individual contributors' attempts to influence articles' contents [33, 56, 64, 65], but these studies did not consider peer production dynamics. Namely, in a large system such as Wikipedia, where participants' contributions tend to be in reference to some existing content (contributed by others), knowing the effects of contributors' actions on others is essential for understanding the system's dynamics. We, thus, extend prior research by investigating the effects of collaborators' behavior on others' actions and studying the system's dynamics.

The finding from the non-cooperative game that a competitive orientation yields content ownership suggests that the benefits associated with content ownership outweigh the effort that is associated with governance-enforcing tax (at least up to a certain limit; see below). This result is rather straightforward, whereas the findings regarding creators and curators is less

intuitive, given that creators are major contributors of content to the article. One may argue that this could be explained by the temporal order of editing: creators add content early on, only to be overwritten by curators' later contributions. However, our temporal bracketing analysis accounts for this possible effect, and we demonstrate that our results hold even when excluding the first year of articles' lives (i.e. the results from Year 2 and onward). How, then, could this result be explained? One line of explanation points to community's effort to ensure neutrality [25, 33]. The "creators" who make large contributions and take ownership of large portions of the article pose a potential threat to the article's neutrality. In response, the community works collectively to fight such manipulations and restores neutrality. In the long run, the result is a more neutral content [30, 84]. Not withstanding the role of community efforts to ensure neutrality, our results may also be explained by paying attention to the profile of those making the large contributions. Prior studies have identified the "creators" [34–37, 85] or "content-oriented" contributors as those whose main interest is to demonstrate their knowledge of the topic, and are much less invested in Wikipedia's internal processes. In contrast, the "curators" (also referred to as "community-oriented" contributors [8]), who engage in refining, shaping, and reorganizing content [86], act as "janitors of knowledge" [87] and end up owning larger portions of articles. That is, at the end of the day, it is the curators who play the leading role in shaping articles' contents. We note that both these lines of explanation highlight the effectiveness of Wikipedia's social processes.

A second powerful result of this study is in demonstrating that that the community's efforts to govern content creation and ensure neutrality, although essential for maintaining a balanced position, should be carefully monitored and kept limited. The reason is that when the "tax" imposed on contributors in terms of complying with NPOV norms, policies and procedures is too high, it outweighs the benefits associated with content ownership, such that contributors stop competing for ownership (and in effect, co-production is stalled). Prior studies have documented the growing bureaucracy of Wikipedia [80], and have suggested that the increasing complexity of Wikipedia's governance structure is deterring newcomers and may eventually lead to the decline of Wikipedia [1]. Our results provide analytical proof of the risk associated with increased governance cost and highlight the need to balance the cost and benefits associated with the governance of online production communities. Furthermore, our analysis indicates that a low level of governance is optimal for ensuring neutrality while maintaining articles' comprehensiveness, suggesting that light-weight governance mechanisms may be optimal. Whereas prior research did suggest that governance levels should be curtailed, the current study adds to the literature by identifying the appropriate levels of governance.

In sum, our study makes important contributions to research on peer production, First we shed light on the dynamics underlying peer production. Second, we show how Wikipedia's governance and in particular, the community's efforts to maintain neutrality, affect contributors' behavior and consequently, the quality of the co-produced articles. Although there have been studies using game theory and network ties to study collaboration in social networks in general [88] and applied to Wikipedia in particular [89], we are not aware of prior research in this area that linked governance to individuals' actions.

Notwithstanding these contributions, our work could be extended in several directions, for example by employing alternative measures for our model's constructs (alternative measures of articles' objectivity through an analysis of Wikipedia's discussion pages) or extending the model to include additional constructs related to contributors (for example: contributors' intent and their psychological ownership, their compliance with Wikipedia's policies, or the phase in an article's life on which they concentrate their edits) or related to the articles (the extent to which articles are controversial or the importance of the knowledge-based product). Future research could also attempt to empirically validate our model's prediction that excessive

levels of governance impede peer production. Finally, a more complex convex cost model (e.g., a quadratic cost model) and its effect on co-production and governance can also be explored.

An additional important contribution of this study is in applying game theory to investigate peer production in Wikipedia. Prior studies have called for the development of new economic models [90] and in particular, the application of game theory to investigate emerging socio-technical systems such as Wikipedia [91]. A few studies have used game-theoretic models to compare the efficiency of closed- and open-source software development regimes [92], investigate the competitive dynamics of the process by which participants provide feedback on each other's work (i.e. rating and reputation systems) [93], and to study co-production in user generated content [94], crowd sourcing [95] and Wikipedia [96]. Our study extends prior work in the area by using game theory to shed light on the role that governance mechanisms play in moderating the competition between contributors to peer production. Our game-theoretic model provides insights into the complexities of cooperation and competition in peer production, while making several simplifying assumptions.

Although our empirical analysis corroborated the game-theoretic model, we did observe small discrepancies between the model's predictions and the empirical data. One possible explanation for these differences is that our model provided only an approximation of Wikipedia's complexities. Future research could help make progress: such work could relax some of our model's assumptions, for example, by distinguishing between different contributors' knowledge and capabilities, modeling learning effects, capturing other important qualities of articles such as accuracy or completeness, and exploring multi-stage games to consider various classes of contributors that differ in their roles and goals. In particular, we foresee three avenues for extending our study. First, in our model a contributor owns his newly-added content thus impacting others' relative ownership. A more sophisticated model could also account for overwriting (or deleting) another contributor's content, thus directly impacting the ownership of a specific other. A challenge for such a model would be to handle the complexity associated with numerous pair-wise relationships. Second, our model could possibly be extended to capture a tighter linkage between the leader-follower game (community governance) and the non-cooperative game (cooperation and competition over content ownership), for example by linking the effectiveness of the community's governance to the aggregate of contributors' governance effort.

Beyond the theoretical contributions and directions for future work, our study's findings also have important implications for custodians of online production communities. The most important insight pertains to the levels of governance that are required. Namely, we find that low levels of governance are sufficient for curbing much of the competition over content ownership, ensuring a relatively even distribution of ownership, while only resulting in a very minor reduction in activity levels, thus having a minimal impact on articles' comprehensiveness. Another important contribution pertains to the importance of peripheral community members that have a less cooperative orientation. In fact, as a consequence of their pursuit of content ownership, their activity levels are increased, such that they add contents to the article, making the article more comprehensive and of higher quality. Hence, communities such as Wikipedia should encourage peripheral participation [97] in addition to their efforts to encourage cooperative participants to take on additional functions and roles.

In conclusion, we believe that game theory can reveal deep insights into the complex dynamics underlying peer production. While game theory, and in particular leader-follower game, has already been employed in the context of allocation of divisible resources [98], the application of such techniques to community-based peer production is novel. Future research directions for game theory in this area include the use of cooperative game theory [99] to study the stability of the production process; applying coalitional game models [100] to analyze

how competing coalitions of contributors emerge, applying resilience and immunity models [101] to investigate how the peer production process performs in cases of deviations from expected behavior, and using evolutionary game theory [102] to study how unexpected behaviors emerge as a result of sequential actions by different contributors. Game theory can also be used to develop models that would account for the relationship between individual's governance work and the community's overall governance level while using algorithm and mechanism design [103] to determine the most effective community governance structures.

## Supporting information

**S1 File. Appendix-1: Nash equilibrium of the non-cooperative game between the contributors.**
(PDF)

**S2 File. Appendix-2: Determining the optimal value of neutrality enforcement, *t*.**
(PDF)

**S1 Data.**
(XLS)

## Acknowledgments

The authors thank the anonymous reviewers for their constructive criticism which greatly enabled us to enhance the results and presentation in the manuscript.

## Author Contributions

**Conceptualization:** Santhanakrishnan Anand, Narayan Mandayam, Oded Nov.

**Formal analysis:** Santhanakrishnan Anand.

**Funding acquisition:** Narayan Mandayam, Oded Nov.

**Investigation:** Oded Nov.

**Methodology:** Santhanakrishnan Anand, Ofer Arazy.

**Project administration:** Narayan Mandayam.

**Supervision:** Narayan Mandayam.

**Visualization:** Santhanakrishnan Anand, Ofer Arazy.

**Writing – original draft:** Santhanakrishnan Anand, Ofer Arazy, Oded Nov.

**Writing – review & editing:** Santhanakrishnan Anand, Ofer Arazy, Oded Nov.

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
