## [Decision Letter · Decision Letter 0]

3 Aug 2021

PONE-D-21-20634

A game-theoretic analysis of competitive editing in Wikipedia: Contributors' effort to influence articles and the community's attempt to ensure neutrality

PLOS ONE

Dear Dr. Anand,

Thank you for submitting your manuscript to PLOS ONE. After careful consideration, we feel that it has merit but does not fully meet PLOS ONE’s publication criteria as it currently stands. Therefore, we invite you to submit a revised version of the manuscript that addresses the points raised during the review process.

At this point two referees have reported. One of them suggests rejection, while the other suggests a major revision. I suggest you to revise and re-submit your manuscript in line with the comments of this latter referee. I want to make clear that my suggestion does not convey any commitment of future acceptance. The real contribution of the current version of the paper seems to be vague and unclear even for the most pro-acceptance referee, which makes me be very cautious at this stage.

We look forward to receiving your revised manuscript.

Kind regards,

Francisco Alvarez Gonzalez

Academic Editor

PLOS ONE

Journal Requirements:

Reviewers' comments:

Reviewer's Responses to Questions

**Comments to the Author**

1. Is the manuscript technically sound, and do the data support the conclusions?

Reviewer #1: Partly

Reviewer #2: Partly

2. Has the statistical analysis been performed appropriately and rigorously? 

Reviewer #1: Yes

Reviewer #2: N/A

3. Have the authors made all data underlying the findings in their manuscript fully available?

Reviewer #1: Yes

Reviewer #2: Yes

4. Is the manuscript presented in an intelligible fashion and written in standard English?

Reviewer #1: Yes

Reviewer #2: Yes

5. Review Comments to the Author

Reviewer #1: The paper uses a game-theoretic model to analyze editing behavior on Wikipedia and in particular, how competition for ownership of an article affects the incentives of individual users to contribute. The paper also analyzes how the Wikipedia’s governance structure influences editing behavior by altering the contribution incentives. The key prediction that emerges from the analysis is that users that make longer edits are less likely to have final ownership over an article (this pattern is verified empirically) and that stricter rules of enforcement on Wikipedia can discourage edits from users that tend to make longer edits (this prediction comes out of the model but is not empirically verified).

Editing dynamics on Wikipedia and related platforms are interesting and understanding how governance rules impact editing is important. Having said that, I have my reservations about the current draft for a few reasons. I do think the prediction about edit length and ownership is driven by an assumption that is baked into the model setup and that I find hard to justify. The second prediction regarding the impact of enforcement I believe is largely driven by the focus solely on competition rather than collaborative elements of the editing process. A study of course always needs to focus on a few important elements. But when thinking about the governance mechanism of a complex system, we might be missing important aspects when just focusing on competition.

1. The one aspect of the model that I found unusual is the distinction between edit size (beta_i) for a given user and edit content x_i for the same user. I don’t quite understand why we need two different concepts here and it seems to me that the key result of the model is driven by this somewhat unusual distinction. The authors shows that content x_i is a negative function of beta_i in equilibrium and therefore higher beta_i users will have lower ownership (ownership is determined only by x_i and independent of edit size beta_i). In some way the main prediction of the model follows mechanically from the assumption about edit utility: users with larger edit sizes beta_i have a higher cost of contributing and therefore contribute less in equilibrium and end up with lower ownership. The natural question then seems to be why the users with long edits do not simply convert to making smaller edits (given that their objective is to maximize their ownership). However, the model does not allow for that and instead takes edit size at the user level as given and exogenous.

2. A related point is that we are taking a static model of competition and comparing it to a dynamic editing process in the data. My sense for why large edits lead to lower ownership is that large edits tend to occur earlier in the life cycle of an article and edit length decreases over time (because contributions are more about polishing up existing content). The later edits by constructions are more likely to survive in the final version of the article.

3. The question of the optimal enforcement mechanism is a very important one, but I am not sure it is easy to answer it without a more complex model of the editing process. If collaborative motives are important for example, then early contributors that make large edits that are later altered might be perfectly happy with this dynamic and in fact might see the curators as providing complementary works to theirs. One finding in this paper is that stronger enforcement will discourage edits from users that make large edits. But this prediction will depend crucially on whether competition for ownership is the primary driving force.

Reviewer #2: This paper provides an analytical model and an empirical analysis to explain the competition between contributors for ownership in crowd-sourced content (as in Wikipedia) and how Wikipedia’s governance of NPOV impacts this. The paper studies an important and interesting topic, and provides intriguing insights on these topics. The combination of the analytical and empirical studies is difficult, but authors make a case for it in this paper. The paper is also generally well written and provides a lot of context and justification for most of the concepts used.

I have a few suggestions for the authors to improve the paper further. First, I think the authors can make their contributions a bit more clear. Even though discussions are made about the different parts, it would be useful to see “what is the point of the paper” somewhere early on, including why a combination of analytical and empirical analysis is appropriate. There should also be a case for why this is interesting, why it matters, and why should the readers care about this. Perhaps some potential managerial results would be useful. Second, some of the assumptions and stated results from the analytical model seem out of place. For example, the model doesn’t really capture heterogeneous \\betas, so I am not sure if the drawn result about creators and curators is valid. Third, the empirical study could improve by some robustness checks, as well as more statistical analysis. Other than these, some of the concepts could be better explained and some assumptions could be better justified. Below I provide my detailed feedback, which I hope can help authors take the paper to the next step in the process, and wish them good luck.

Major Comments

- Is NPOV the same as equal contribution? Does Wikipedia generally prefers many contributors to fewer, thereby motivating smaller edits? Authors state without enough evidence that often the reason for a lack of neutrality is that one contributor takes substantial ownership. It seems to me that having a NPOV is not the same as having many contributors for an article: one contributor may be able to provide a NPOV on a subject. Therefore, it seems to me that the model should capture these different concepts in a more nuanced way. In other words, the governance does not directly impact the number of contributors, but this is more of an indirect impact, as NPOV may require a more diverse set of contributors. I understand that the analytical model cannot capture the whole reality, however, these two concepts are focal to the analysis. Alternatively, authors can provide additional evidence that shows a direct connection between the two.

- Another concept related to the point above is that different contributors may have similar POVs, thereby, even as the number of contributors increases, if they all have similar ideas, then NPOV would not be achieved.

- Do contributors really try to maximize their ownership? Or are they trying to maximize the informativeness of the articles? Is this concept more prevalent in articles which are amenable to different points of view, for example in case of more controversial articles? Perhaps some analysis of the data on different types of articles (controversial vs non- controversial) would provide additional justification about the competition on ownership.

- Do the contributors need to put effort in coordination and governance work and sit on committees for Wikipedia? This is somewhat at odds with the assumption that policies don’t change over time. Related to this, from a modeling point of view, L is unnecessary, as it is always multiplied by \\beta (the new \\beta is L \\beta). Also, t could be included in \\beta, therefore it is also unnecessary. Basically, there is a fixed cost and a variable cost to editing, and only the total cost is important, as the individual parts do not act differently from one another. Additionally, it is not clear why the contributor incurs the fixed cost for every edit they do? Wouldn’t they become familiar with the platform at some point and thereby not concur the fixed cost? The authors don’t really provide much justification or example for this type of cost.

- Authors mention quadratic cost, is it possible to get any results with a quadratic function? Such a function makes a lot more sense: it is easy to add a few sentences to an article, but it gets prohibitively hard as they keep editing and adding to an article. (Though I can see the model getting stuck with such a structure.)

- It may be better to provide the results in a more formal way, e.g. in a lemma or proposition.

- Given the topic, heterogeneous user abilities (incurring different costs for editing) seems critical? I understand that this may not be possible to solve for the case of an arbitrary number of contributors, but perhaps the robustness can be shown with a few contributors that are heterogeneous? Next item also considers this issue.

- The second result from equation 6 on page 11 is not accurate: given that there is only one type of user in the model (all \\betas are the same), one cannot compare the curators and creators at the same time. The model captures either the case where all users are creators or where all contributors are curators, not where both types of users are present. In other words, it compares the case where users on average create more content to the case where on average they create less content. I believe that there is a difference between these two, and authors need to clarify this. Moreover, the connection of the model with the empirical analysis should also be reconsidered, given that the empirical results are based on the difference between contributors’ \\betas.

- In the Wikipedia’s decision in the game, does it have any other goals than NPOV? It may be interesting to see if there are any trade-offs, e.g. in size or number of articles (which may be hindered as governance increases).

- The 50% threshold for owning an article seems somewhat arbitrary, are the results robust to other measures of ownership (if there are any other measures for this in the literature for example)? It would also be nice to see some stats on how much of the articles are actually owned by a contributor (i.e. larger than the threshold).

- Rather than the figures for the empirical part, perhaps authors can statistically test their analytical model hypotheses, e.g. the impact of average article size on ownership?

- Given the timing effects in the empirical analysis, I am confused about something: isn’t the result simply because those who have already made lots of edits have not much left to say, so obviously they would make smaller edits? Is \\beta calculated in the context of only the sample of articles or is it their overall value?

- What are the implications of the findings for Wikipedia? How does this impact the quality of articles, number of articles, etc? It would be nice to discuss some of these implications for Wikipedia and other similar platforms.

Minor issues

- There are some issues with the writing:

o “loose ownership of the article to those…” should be “lose…”. This error is made in other parts as well. Loose means not firmly attached, lose means to not have something (e.g. lose ownership)

o “and their resulting content ownership” is repeated twice on page 16.

- There are some vague concepts in the beginning of the paper, for example, what is Wikipedia’s refactoring process? It would be great to have someone proof-read so that all concepts are explained fully.

6. PLOS authors have the option to publish the peer review history of their article (what does this mean?). If published, this will include your full peer review and any attached files.

Reviewer #1: No

Reviewer #2: No

---

## [Author Response · Author response to Decision Letter 0]

2 Mar 2022

We have addressed all the comments. We have prepared a detailed Rebuttal Letter. All our responses are included in the rebuttal letter

---

## [Decision Letter · Decision Letter 1]

21 Jul 2022

PONE-D-21-20634R1A Game-Theoretic Analysis of Wikipedia’s Peer Production: the Interplay between Community’s Governance and Contributors’ InteractionsPLOS ONE

Dear Dr. Anand,

Thank you for submitting your manuscript to PLOS ONE. After careful consideration, we feel that it has merit but does not fully meet PLOS ONE’s publication criteria as it currently stands. Therefore, we invite you to submit a revised version of the manuscript that addresses the points raised during the review process.

We look forward to receiving your revised manuscript.

Kind regards,

Karthikeyan Rajagopal

Academic Editor

PLOS ONE

Additional Editor Comments (if provided):

Reviewers have now requested some revisions to the manuscript.

Reviewers' comments:

Reviewer's Responses to Questions

**Comments to the Author**

1. If the authors have adequately addressed your comments raised in a previous round of review and you feel that this manuscript is now acceptable for publication, you may indicate that here to bypass the “Comments to the Author” section, enter your conflict of interest statement in the “Confidential to Editor” section, and submit your "Accept" recommendation.

Reviewer #1: All comments have been addressed

Reviewer #2: (No Response)

2. Is the manuscript technically sound, and do the data support the conclusions?

Reviewer #1: Yes

Reviewer #2: Yes

3. Has the statistical analysis been performed appropriately and rigorously? 

Reviewer #1: Yes

Reviewer #2: Yes

4. Have the authors made all data underlying the findings in their manuscript fully available?

Reviewer #1: Yes

Reviewer #2: Yes

5. Is the manuscript presented in an intelligible fashion and written in standard English?

Reviewer #1: Yes

Reviewer #2: Yes

6. Review Comments to the Author

Reviewer #1: (No Response)

Reviewer #2: Summary

I believe that the authors are moving towards the correct direction in this new revision. I believe that the representation of contributors using the spectrum for the curator versus creator contributors as well as their competitive and cooperative urges is a nice addition and has the potential to provide some interesting insights. The authors have also done a good job introducing some of the concepts in more detail and clarity. There are still however, a few issues that need further explanation or clarification in the model. Particularly the relationship between the number of edits, x, and the user type in terms of creator versus curator, beta, is a confusing. In addition to this there are many other clarifications that might be needed as I explained below, and there is room to further elaborate on the findings. There are also many grammatical and writing issues feature I believe could be resolved by having a professional editor proofread the manuscript.

Major Comments

-I wasn't sure why cost of making an edit depends on the contributors activity profile brta_i. Considering the definition of the fractional ownership in c_i which is based on, x_i, this should be a standardized measure of contribution. For example in equation 1 the fractional content ownership is defined as the relative measure of contributed content over total content, which would be incorrect if the size of different x_i’s are different for different types of contributors. In other words, if a creator writes more in one edit compared to a curator, they should still own more of the article. It does not seem to me that the number of edits should be used to calculate the fractions but instead the amount of contribution. Therefore, I'm not sure if the effort involved in a single edit should depend on the type of the user, beta_i. Should beta_i be represented in how much content the user produces instead? In other words, would higher beta users have higher x_i?

-Related to the above comment, understanding this was harder as it is not clear weather creators have low beta or a high beta. From lines 576-579, it seems that data is the size of the edit. This is a little bit at odds with the definition of beta and x_i. Moreover, using this definition, I'm a bit confused about why the cost or effort for a single edit is defined as it is. It seems to me that the creators’ effort for creating the same amount of information would be lower than curators, in other words, it seems to me that the reverse of what the authors propose makes more sense. Overall it seems to me that the authors should think about the concepts of beta and x a bit more, as they currently seem inconsistent. As a suggestion, I wonder if creation versus curation can be considered just in terms of heterogeneity for the amount of content created, that is, x_i, removing the need for beta.

-It is not clear what f_i is when it is introduced first. Moreover, rather than identifying this cost as being applicable only for the cooperative contributors in line 493, it seems to me that the authors should represent this as being higher for cooperative contributors. This stems from the fact that this cost depends on w in a continuous manner. Additionally this cost does not seem to be impacting any of the results. I believe it would be nice if the authors explain and clarify this in the paper.

-The same idea with the continuous w_i applies for the governance tax when discussing its effect.

-I Also made this point in the last round about the convex cost function. Linear is not convex when talking about cost. The paper you refer to is in the context of linear programming, not game theory, where a convex cost function is different from linear. I let the AE decide whether it is necessary to actually try a convex cost function. It doesn’t seem to me that this would impact the main results. If you decide to leave out the convex cost function you can just focus on what is in the paper, but there's no need to mention convex cost function and it being outside of the scope of this paper.

-I believe that even if the findings are not used as intermediate findings to prove further results (which I think does take place going from equation 4 to 6), it is still useful to provide the results more formally. Perhaps you can use a theorem instead for some of the results, for example those coming from equation 4, the optimal amount of content created, and equation 6, fractional content ownership. Looking at some similar papers that consider cooperative game theory might also help you better represent the equilibrium results.

-When explaining the impact of t on equation 4 starting in line 563, I don't see a second order quadratic factor in the denominator for t. Perhaps comparative statics might be a better way to show this.

-I found the results pertaining to w and beta to be interesting. I'm wondering if there is a better way to show this by considering the two dimensions of user heterogeneity. In other words the authors can draw a plot with these two dimensions and then characterize the different types of contributors within that two-dimensional space to better demonstrate their results.

-Is it really the case that the type of users has no correlation with the amount of edits? Do privileged members contribute the same amount as the other users? Even though the authors provide some literature on this it would be nice to see some actual evidence. Even if the there is some correlation between these, I believe that the results would continue to hold, but they just need to discuss this further.

-I believe that the piecewise nature of plot in figure 3 need additional explanation. What are the jumps that happen in these plots at around indexes 6, 8 and 9?

-The non-human editors need additional information. What are the implications of this? This is a bit out of the blue.

-It wasn't clear to me how the level of governance t was measured in the empirical part.

-In the introduction before getting to the results the definition of governance is a little bit vague. It would help if the authors can add some examples of what governance actually means.

Minor Comments

-There are still some issues with the writing:

oIn line 127, there needs to be a “to”

oIn line 421, there is a missing space after comma.

oMissing dot in line 466 and 714.

oIn line 462, there's an extra space after the first parenthesis.

oIn line 634, should this be beta_i instead of beta?

oIn line 493, contributirs is incorrect.

-Use of parentheses () For both functions and multiplications for example in equation two can be confusing. I let the authors decide what they want to do with this.

-When first introducing beta in 460 it would be nice to provide a definition of the continuum, that is what does a low or a high beta mean. I had a hard time finding this.

7. PLOS authors have the option to publish the peer review history of their article (what does this mean?). If published, this will include your full peer review and any attached files.

Reviewer #1: No

Reviewer #2: No

---

## [Author Response · Author response to Decision Letter 1]

21 Sep 2022

We have now attached a rebuttal letter with detailed responses to the reviewers' comments. We have incporporated changes in the revised manuscript and attached the data file used in our analysis

---

## [Decision Letter · Decision Letter 2]

15 Nov 2022

PONE-D-21-20634R2A Game-Theoretic Analysis of Wikipedia’s Peer Production: the Interplay between Community’s Governance and Contributors’ InteractionsPLOS ONE

Dear Dr. Anand,

Thank you for submitting your manuscript to PLOS ONE. After careful consideration, we feel that it has merit but does not fully meet PLOS ONE’s publication criteria as it currently stands. Therefore, we invite you to submit a revised version of the manuscript that addresses the points raised during the review process.

We look forward to receiving your revised manuscript.

Kind regards,

Karthikeyan Rajagopal

Academic Editor

PLOS ONE

Journal Requirements:

Reviewers' comments:

Reviewer's Responses to Questions

**Comments to the Author**

1. If the authors have adequately addressed your comments raised in a previous round of review and you feel that this manuscript is now acceptable for publication, you may indicate that here to bypass the “Comments to the Author” section, enter your conflict of interest statement in the “Confidential to Editor” section, and submit your "Accept" recommendation.

Reviewer #2: All comments have been addressed

2. Is the manuscript technically sound, and do the data support the conclusions?

Reviewer #2: Yes

3. Has the statistical analysis been performed appropriately and rigorously? 

Reviewer #2: Yes

4. Have the authors made all data underlying the findings in their manuscript fully available?

Reviewer #2: (No Response)

5. Is the manuscript presented in an intelligible fashion and written in standard English?

Reviewer #2: Yes

6. Review Comments to the Author

Reviewer #2: The authors have addressed almost all of my prior comments in this revision. I want to commend the authors in their efforts to update the model, as well as to re-run all of the empirical analyses. I am generally happy about the state of the paper at this point, and just have a few minor edits plus some editorial suggestions. I hope the authors find them useful. Given that these suggestions are not fundamental to the paper, I recommend a minor revision at this point. Below are my detailed comments.

-Does the quality (sum of all x_i) need a multiplier so that the relative magnitude of quality and the fractional ownership can be better accounted for? I am worried that the way the utility function is modeled in (2), the quality may have an outsized impact on the overall quality compared to the fractional ownership, given that perhaps the magnitude of the argument for quality is going to be significantly larger than the fractional ownership argument. I am not sure if this impacts anything, and perhaps it doesn’t, but just a quick confirmation that this is not driving the results would help.

-The asymptotic results in lines 574-587 states that the result are in “the presence of Wikipedia's governance to ensure neutrality”, however, the result from Theorem 4 seems to hold irrespective of t. The authors need to clarify that this is just because of the costs and benefits through the utility function other than t that those users with low edit sizes and high competitiveness end up owning more share. It might be nice to add a bit of intuition here based on not just the asymptotic case, but also the general case in Theorem 3 on why this happens.

-I think I didn’t properly explain what I meant before when I asked the authors to take a look at the plot that has w and \\beta on two dimensions. I was asking for the plot from the game-theoretic model on these two. Basically, for a given parameter set, a plot can be drawn to show for example that those users with w/\\beta < E[W/\\Beta] own a non-significant portion of the content, whereas those with w/\\beta > E[W/\\Beta] do not own any of the content per Theorem 4 (or perhaps find a way to discuss the findings in Theorem 3?).

-In the literature review section, it might be useful to further clarify the contribution to each of the three sub-fields and quickly compare the stated literature to the current paper on the aspects that are important and relevant.

-When introduction \\beta_i, clarify the continuum for it: does high \\beta mean creator or curator? This is formally clarified much later on line 567.

-The y-axis on Figure 4 goes from 0 to 10^0, perhaps better to note it as 0 to 1?

-Research question may be better provided not in a section of its own, though this is subjective and I let the authors decide whether to change anything about this.

-Writing: I found a couple of issues, you may want to have another proof-reading before the final version

oOn line 430, the “a” before “either” should be removed and there is an extra space after the “(“.

oOn line 574, it should be “who ARE creators of content”.

7. PLOS authors have the option to publish the peer review history of their article (what does this mean?). If published, this will include your full peer review and any attached files.

Reviewer #2: No

---

## [Author Response · Author response to Decision Letter 2]

13 Jan 2023

We thank the reviewers for their comments. We have responded to the comments and revised the manuscript in accordance to the reviewers' comments. We have uploaded our responses as a separate document

---

## [Decision Letter · Decision Letter 3]

31 Jan 2023

A Game-Theoretic Analysis of Wikipedia’s Peer Production: the Interplay between Community’s Governance and Contributors’ Interactions

PONE-D-21-20634R3

Dear Dr. Anand,

We’re pleased to inform you that your manuscript has been judged scientifically suitable for publication and will be formally accepted for publication once it meets all outstanding technical requirements.

Kind regards,

Karthikeyan Rajagopal

Academic Editor

PLOS ONE

Additional Editor Comments (optional):

Reviewers' comments:

Reviewer's Responses to Questions

**Comments to the Author**

1. If the authors have adequately addressed your comments raised in a previous round of review and you feel that this manuscript is now acceptable for publication, you may indicate that here to bypass the “Comments to the Author” section, enter your conflict of interest statement in the “Confidential to Editor” section, and submit your "Accept" recommendation.

Reviewer #2: All comments have been addressed

2. Is the manuscript technically sound, and do the data support the conclusions?

Reviewer #2: Yes

3. Has the statistical analysis been performed appropriately and rigorously? 

Reviewer #2: Yes

4. Have the authors made all data underlying the findings in their manuscript fully available?

Reviewer #2: (No Response)

5. Is the manuscript presented in an intelligible fashion and written in standard English?

Reviewer #2: (No Response)

6. Review Comments to the Author

Reviewer #2: The authors have addressed all of my comments from the previous round. I want to congratulate them for their efforts and the quality of the paper.

7. PLOS authors have the option to publish the peer review history of their article (what does this mean?). If published, this will include your full peer review and any attached files.

Reviewer #2: No

---

## [Editor Report · Acceptance letter]

20 Feb 2023

PONE-D-21-20634R3 

A Game-Theoretic Analysis of Wikipedia’s Peer Production: The Interplay between Community’s Governance and Contributors’ Interactions 

Dear Dr. Anand:

I'm pleased to inform you that your manuscript has been deemed suitable for publication in PLOS ONE. Congratulations! Your manuscript is now with our production department. 

Kind regards, 

on behalf of

Dr. Karthikeyan Rajagopal 

Academic Editor

PLOS ONE